# Fetal, neonatal, and infant outcomes associated with maternal Zika virus infection during pregnancy: A systematic review and meta-analysis

Marlos Melo Martins[1]*, Antonio José Ledo Alves da Cunha[2], Jaqueline Rodrigues Robaina[3], Carlos Eduardo Raymundo[3], Arnaldo Prata Barbosa[3], Roberto de Andrade Medronho[4]

1 Department of Pediatrics, Martagão Gesteira Institute of Childcare and Pediatrics, Federal University of Rio de Janeiro, Rio de Janeiro, Brazil, 2 Department of Pediatrics, School of Medicine, Federal University of Rio de Janeiro, Rio de Janeiro, Brazil, 3 Department of Pediatrics, Instituto D'Or de Pesquisa e Ensino (IDOR), Rio de Janeiro, Brazil, 4 Department of Epidemiology and Public Health, School of Medicine, Federal University of Rio de Janeiro, Rio de Janeiro, Brazil

☯ These authors contributed equally to this work.
* marlosmartins@globo.com

**Data Availability Statement:** All relevant data are within the paper and its Supporting Information files.

## Abstract

The occurrence of fetal and neonatal disorders in pregnant women with Zika virus infection in the literature is not consistent. This study aims to estimate the prevalence rate of these disorders in fetuses/neonates of pregnant women with confirmed or probable infection by Zika virus. A systematic review with meta-analysis was conducted in November 2020. Cohort studies that contained primary data on the prevalence of unfavorable outcomes in fetuses or neonates of women with confirmed or probable Zika virus infection during pregnancy were included. A total of 21 cohort studies were included, with a total of 35,568 pregnant women. The meta-analysis showed that central nervous system abnormalities had the highest prevalence ratio of 0.06 (95% CI 0.03–0.09). Intracranial calcifications had a prevalence ratio of 0.01 (95% CI 0.01–0.02), and ventriculomegaly 0.01 (95% CI 0.01–0.02). The prevalence ratio of microcephaly was 0.03 (95% CI 0.02–0.05), fetal loss (miscarriage and stillbirth) was 0.04 (95% CI 0.02–0.06), Small for Gestational Age was 0.04 (95% CI 0.00–0,09), Low Birth Weight was 0.05 (95% CI 0.03–0.08) and Prematurity was 0.07 (95% CI 0.04–0.10). The positivity in RT-PCR for ZIKV performed in neonates born to infected mothers during pregnancy was 0.25 (95% CI 0.06–0.44). We also performed the meta-analysis of meta-analysis for microcephaly with the prevalence ratios from other two previously systematic reviews: 0.03 (95% CI 0.00–0.25). Our results contribute to measuring the impact of Zika virus infection during pregnancy on children's health. The continuous knowledge of this magnitude is essential for the implementation development of health initiatives and programs, in addition to promoting disease prevention, especially in the development of a vaccine for Zika virus. PROSPERO protocol registration: http://www.crd.york.ac.uk/PROSPERO/display_record.php?ID=CRD42019125543.

**Funding:** No funding for our study.

**Competing interests:** NO authors have competing interests.

## Introduction

In October 2015, in the Brazilian state of Pernambuco, an increase in the number of cases of microcephaly was detected, triggering the first field investigations into its causes. At the same time, other states in other regions of Brazil detected an increase in the number of neonates with congenital microcephaly and other clinical characteristics similar to those described initially in the state of Pernambuco [1].

Since then, evidence has accumulated in favor of the association between Zika virus (ZIKV) infection in pregnancy and congenital microcephaly and other neurological and clinical abnormalities in fetuses/neonates [2,3]. The detection of genetic material successively strengthened this association of the virus in different biological materials collected from affected pregnant women and fetuses/neonates. ZIKV was detected in brain tissue and placentae of neonates and dead fetuses [4], in the amniotic fluid of pregnant women with fetuses presenting intrauterine microcephaly [5] and in the fetal brain tissue, after the termination of pregnancy of a Slovenian pregnant woman who had exanthematous febrile illness in the first trimester when she lived in Brazil [6]. Subsequently, the causal relationship between ZIKV infection during pregnancy and the occurrence of congenital anomalies in the fetus was recognized internationally [7–9].

ZIKV belongs to the flavivirus genus of the family Flaviviridae. The family Flaviviridae (from the Latin *flavus*, which means yellow, due to jaundice typically caused by the yellow fever virus) is composed of three genera: Flavivirus, Pestivirus and Hepacivirus. The Flavivirus genus comprises about 39 species, including arboviruses [10]. Currently, two strains of ZIKV are recognized: African and Asian [11].

Despite all the rapid knowledge acquired since the beginning of the ZIKV outbreak in Brazil, there are still many knowledge gaps to be filled, reinforcing the importance, relevance, and timeliness of further studies on the infection and its consequences. The prevalence of perinatal outcomes in pregnant women with proven ZIKV infection is described in the literature by some cohort studies or case series, but with variable prevalence rates. There is a lack of studies comparing these rates between infected and not infected pregnant women with the Zika virus. Thus, our main objective is to estimate the prevalence rate of perinatal outcomes possibly related to ZIKV infection in pregnant women such as microcephaly, central nervous system (CNS) abnormalities, miscarriage, stillbirth, prematurity, small for gestational age (SGA), low birth weight (LBW), and results from reverse transcription polymerase chain reaction (RT-PCR) for ZIKV performed in neonates, through a systematic review and meta-analysis. Thus, expanding knowledge about how ZIKV can interfere in fetal development and its outcomes may aid in the development of different health initiatives and programs, especially those associated with perinatal care, besides promoting the disease prevention.

## Materials and methods

A systematic review in compliance with the PRISMA (Preferred Reporting Items for Systematic Reviews and Meta-analysis) framework was carried out [12]. The articles selection process was conducted out in four stages:

1. identification of the articles by searching the different databases;

2. selection; during this phase, duplicate articles were excluded and the selection of the remaining articles was conducted by the title and abstract screening;

3. eligibility; full reading of the articles selected in the previous phase, excluding those that did not meet the pre-established eligibility criteria and;

4. inclusion of eligible articles in the systematic review.

The systematic review was performed in November 2020. The databases used were Medline / PubMed, SciELO (Scientific Eletronic Library Online), Lilacs (Latin-American and Caribbean System on Health Sciences Information), Web of Science, Scopus, Cochrane Library, Portal CAPES (Coordination for Higher Education Staff Development) and CINAHL (Cumulative Index of Nursing and Allied Health Literature). Other databases were also used, such as Scisearch, Australasian Medical Index, database of theses and dissertations from USP (University of São Paulo), PUC (Pontifical Catholic University) and CAPES, in addition to the ProQuest Dissertations Theses Database, BMC Central Proceedings and BMC Meeting Abstracts. For grey literature, Google Scholar was used.

Descriptors were chosen according to the DeCS (Health Sciences Descriptors) and MeSH (Medical Subject Headings). The uncontrolled vocabulary was also used, which consisted of text words, acronyms, related terms, keywords and spelling variations, in addition to the "entry terms" indexed to the descriptors in MeSH. Descriptors and uncontrolled vocabulary in English, Portuguese and Spanish were used, applying the Boolean operators OR and AND to combine the terms in the databases. The descriptors used were "Zika Virus" OR "Zika Virus Infection" OR "ZikV" OR "Virus, Zika" OR "Infection, Zika Virus" OR "Virus Infection, Zika" OR "ZikV Infection" OR "Fever, Zika" OR "Zika Virus Disease" OR "Disease, Zika Virus" OR "Virus Disease, Zika" OR "Zika Fever" AND "Microcephaly" OR "Congenital Abnormalities" OR "Nervous System Diseases" OR "Neurologic Manifestations" OR "Microcephalies" OR "Congenital Abnormality" OR "Congenital Defects" OR "Birth Defects" OR "CNS Disease" OR "CNS Diseases" AND "Epidemiology" OR "Prevalence" OR "Incidence" OR "Cohort" OR "Frequency" OR "Occurrence". The full electronic search strategy used to identify studies with all search terms and limits for all databases is described in S1 Table.

The period of publication was from January 2015 to November 2020. Articles were found in English, Spanish and Portuguese. Only cohort studies containing primary data on the prevalence of unfavorable outcomes in fetuses or neonates of women with confirmed or probable ZIKV infection during pregnancy were included. Letters to the editor, case series, ecological studies, case-control studies, cross-sectional studies, research protocols, non-systematic reviews and studies of epidemiological models were excluded.

For the article selection stage, the Mendeley Reference Manager software, version 1.19.4, was used as a reference manager for screening, with the initial exclusion of duplicates. A double pair of reviewers was used for the title and abstract screening, independently. Articles that did not meet the eligibility criteria were excluded. Disagreements between reviewers were resolved by consensus between the two reviewers or by using a third reviewer. The agreement between reviewers was measured using Cohen's Kappa statistic [13].

In the eligibility stage, a standardized eligibility assessment form was used previously prepared with the inclusion and exclusion criteria. For the proper refinement of the articles, the following inclusion criteria were defined: cohort studies (prospective or retrospective), pregnant women with positive PCR for Zika virus or positive IgM for Zika virus with the plaque reduction neutralization test (PRNT), description of fetal or neonatal outcomes as miscarriage (< 20 weeks of gestational age), stillbirth (= or > 20 weeks of gestational age), congenital microcephaly (head circumference at birth below at least two standard deviations from the mean for gestational age and sex), central nervous system (CNS) abnormality (detected by fetal or neonatal imaging), small for gestational age (SGA), low birth weight (LBW), prematurity (PMT) and neonatal RT-PCR ZIKV infection test performed in serum, urine or cerebrospinal fluid. Eyes and congenital ear abnormalities were also searched. Exclusion criteria were qualitative studies, non systematic review articles, editorials, letters to the editor, book

chapters, non-complete articles and articles that did not present data on the occurrence of unfavorable neonatal outcomes in pregnant women with confirmed or probable ZIKV infection.

The reasons for exclusion of the articles after reading the full text are elucidated in the flow of the selection of articles in the systematic review. The Strengthening the Reporting of Observational Studies in Epidemiology (STROBE) statement tool was used to critically appraise the included observational studies [14]. We also appraised the quality of each study according to criteria in the S2 Table, adapted from Joanna Briggs Institute criteria for assessing incidence/prevalence studies [15]. For each criterion, the studies were classified as having met the criteria or not in terms of providing sufficient or insufficient information to judge.

Two reviewers extracted the data from selected articles, including authors' names, published journal, year of publication, study design, study location, study period, inclusion and exclusion criteria, number of exposed and unexposed pregnant women to ZIKV infection, outcomes studied and statistical methods used. In those studies which compared pregnant women with and without ZIKV infection, we extracted all other characteristics that could represent possible confounding and interaction factors. A meta-analysis of the proportions with a 95% CI was performed for the outcomes described in at least three articles: congenital microcephaly, congenital neurological abnormalities, miscarriage, stillbirth, SGA, LBW, PMT and neonatal RT-PCR for ZIKV infection test performed in any organic fluid. Among the congenital neurological abnormalities, we also performed a meta-analysis of proportions with a 95% confidence interval for brain calcifications and ventriculomegaly. We used the R Studio program, version 1.1.453, for the meta-analysis. We used a binary random effects model, assuming that the proportion of the congenital outcomes in infants/fetuses of ZIKV-infected mothers varies across populations. The meta-analysis was performed to consider the heterogeneity between articles. Heterogeneity among articles was tested by using the Cochran Q test with a significance level of 0.10 that informs about the presence versus the absence of heterogeneity, and was qualified by using the $I^2$ statistic that quantifies the degree of heterogeneity among studies ($I^2$ 0–25% non-important heterogeneity, 25–50% moderate heterogeneity and> 50% considered high) [16].

## Results

### Selection of articles

The initial database and bibliography search resulted in a total of 3,914 records (2,711 records in the electronic databases and 1,203 in the grey literature). The elimination of duplicates was performed by the Mendeley Reference Manager, resulting in 1,341 articles. The title and abstract screening, independently conducted by the two reviewers, resulted in 59 articles. The agreement between the reviewers, in the screening stage based on the title and abstract in the screening stage, was measured with a Cohen's Kappa statistic: 0.935 (95% CI 0.883 to 0.987). Cohen suggested the Kappa result be interpreted as follows: values 0–0.20 as indicating no agreement, 0.21–0.39 as a minimal level of agreement, 0.40–0.59 as weak, 0.60–0.79 as moderate, 0.80–0.90 as strong and above 0.90 as almost perfect [17]. Of the 59 complete articles selected, the same pair of reviewers, also independently, selected 23 articles for systematic review and meta-analysis. In this phase, the agreement between reviewers was also measured using Cohen's Kappa statistic, with a value found of 0.754 (95% CI 0.569 to 0.938). Despite this research, none of the included studies in the systematic review was from grey literature. The reasons for excluding the other 36 articles were: three letters to the editor, four case series, five case-control studies, 16 cohorts without eligibility criteria, one study protocol, one non-systematic review and one model study. Five duplicate articles were also identified. Two

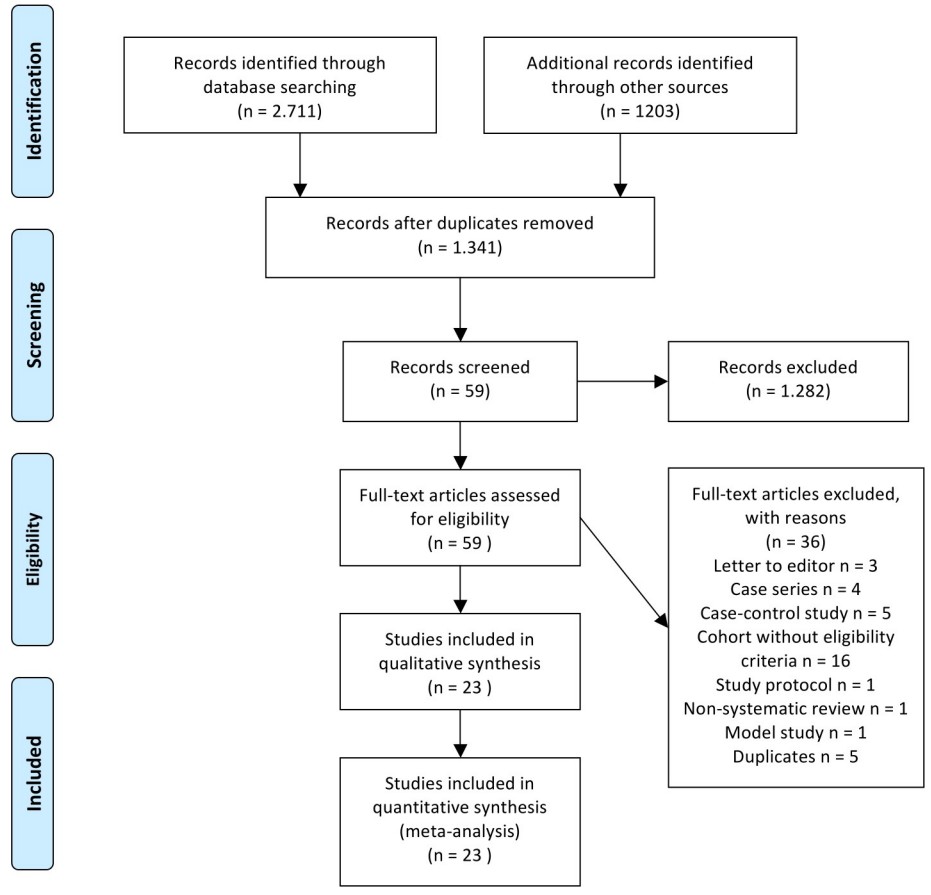

**Fig 1. Articles selection flow in the systematic review.**

systematic reviews were identified, which were also included. Fig 1 summarizes the articles selection flow for the systematic review.

Table 1 summarizes the main characteristics of the 23 studies included in the systematic review [18–40].

## Study characteristics and prevalence of perinatal/neonatal outcomes

The systematic analysis of the 23 selected articles (21 cohorts and two systematic reviews) resulted in a total of 35,568 pregnant women with confirmed or probable ZIKV infection in the cohort studies. The variation in the number of pregnant women in the studies ranged from 28 [19] to 19,963 [39]. The overall studies were conducted in the Americas: Brazil [25,26,29,33,34,38], Puerto Rico [18], United States of America (USA) [19,21,22,24,35,37,40], Peru [30], French territories of the Americas: French Guiana, Guadeloupe and Martinique [20,36] and Colombia [23,27,28,39]. Quality scores for the studies are available in S3 Table. Of 21 cohort studies, nine met all five quality criteria, nine met four, two met three, one met two, and none met only one or did not meet any.

The criteria for ZIKV infection followed that recommended by the Center for Disease Control and Prevention (CDC) [41]: positive RT-PCR in serum or urine or IgM for ZIKV by the enzyme-linked immunosorbent assay (ELISA) method, including the PRNT for the differential diagnosis with other arboviruses. In 11 cohorts, only pregnant women with positive

**Table 1. Characteristics of the studies included in the systematic review.**

| Study ID | Country | Design | Period of study | Number of infected pregnant women | Microcephaly | CNS abnormalities | Intracranial calcifications | Ventriculomegaly | Miscarriage | Stillbirth | SGA | LBW | Prematurity | Neonatal RT-PCR positive (Newborn/RT-PCR positive) |
|---|---|---|---|---|---|---|---|---|---|---|---|---|---|---|
| Adam et al. 2006 [18] | Puerto Rico | Cohort in retrospective | Nov 15- Jul16 | 672 | 0 | 0 | --- | --- | 2 (0.3%) | --- | --- | --- | --- | --- |
| Adhikari et al. 2017 [19] | USA | Cohort in prospective | Mar16-Oct16 | 28 | 0 | 1 (3.57%) | --- | --- | --- | 0 | --- | 2 (7.84%) | 2 (7.84%) | --- |
| Asplicueta-Gloa et al. 2017 [39] | Peru | Cohort in retrospective | Jan16-Dec16 | 38 | 0 | --- | --- | --- | --- | 0 | 2 (5.26%) | --- | --- | --- |
| Brasil et al. 2016 [14] | Brazil | Cohort in prospective | Sep15-May16 | 117 | 4 (3.4%) | 4 (3.4%) | --- | --- | 6 (4.8%) | 3 (2.4%) | 2 (1.7%) | --- | --- | --- |
| Hoff et al. 2017 [15] | USA | Cohort in retrospective | Jan16-Jul17 | 185 | 10 (5.40%) | --- | --- | --- | 3 (1.62%) | 0 | --- | --- | 17 (14.5%) | --- |
| Hoen et al. 2018 [16] | French Guiana, Guadalupe, Martinique | Cohort in prospective | Mar16-Nov16 | 546 | 32 (5.86%) | 13 (2.38%) | 8 (1.46%) | 8 (1.46%) | 11 (2%) | 6 (1.09%) | --- | --- | --- | --- |
| Honein et al. 2017 [17] | USA | Cohort in retrospective | Dec15-Nov16 | 442 | 18 (4.07%) | 18 (4.07%) | 11 (2.49%) | --- | 47 (10.6%) | 0 | --- | --- | --- | 28/6 (21.4%) |
| Joao et al. 2018 [38] | Brazil | Cohort in prospective | Jan15-Aug16 | 34 | 3 (6.82%) | 2 (5.88%) | 2 (5.88%) | 1 (2.94%) | --- | --- | --- | --- | --- | --- |
| Mendez et al. 2017 [39] | Colombia | Cohort in retrospective | Oct15-Jan17 | 19983 | 730 (3.55%) | --- | --- | --- | --- | --- | --- | --- | --- | --- |
| Muller et al. 2019 [40] | USA | Cohort in prospective | Jun16-Jun17 | 82 | 1 (1.21%) | 1 (1.21%) | 1 (1.21%) | --- | 1 (1.21%) | --- | --- | --- | --- | --- |
| Pomar et al. 2017 [20] | French Guiana, Guadalupe, Martinique | Cohort in prospective | Jan15-Jul16 | 301 | 6 (1.99%) | 27 (8.97%) | --- | 5 (1.66%) | 9 (2.99%) | 0 | --- | --- | --- | --- |
| Reynolds et al. 2017 [21] | USA | Cohort in retrospective | Jan15-Dec16 | 972 | --- | 51 (5.24%) | --- | --- | 77 (7.92%) | --- | --- | --- | --- | 58/594 (10%) |
| Rice et al. 2008 [12] | USA | Cohort in retrospective | Feb17-Jan18 | 1,450 | 84 (5.79%) | 87 (11.24%) | --- | --- | 155 (3.58%) | --- | --- | --- | --- | 84/132 (1.8%) |
| Rodriguez-Morales et al. 2018 [23] | Colombia | Cohort in retrospective | Jan16-Dec16 | 86 | 2 (2.32%) | 0 | --- | --- | --- | --- | --- | 1 (1.16%) | 2 (2.32%) | --- |
| Shapiro-Mendoza et al. 2017 [24] | USA | Cohort in retrospective | Jan16-Apr17 | 2,549 | --- | 122 (4.78%) | 1 (0.85%) | 1 (0.85%) | 85 (3.33%) | --- | --- | --- | --- | --- |
| Sousa et al. 2020 [15] | Brazil | Cohort in retrospective | Jan 15- Jun17 | 117 | 13 (11.1%) | 14 (12%) | 1 (0.85%) | 1 (0.85%) | 3 (2.56%) | 1 (0.85%) | --- | --- | 12 (10.25%) | 13/5 (18.46%) |
| Sanchez-Clemente et al. 2020 [26] | Brazil | Cohort in prospective | Mar16-Aug17 | 44 | 2 (4.3%) | --- | --- | --- | 0 | --- | 4 (9.09%) | 4 (9.09%) | 4 (9.09%) | --- |
| Ocampo-Grause et al. 2020 [27] | Colombia | Cohort in retrospective | Oct15-Jul16 | 1758 | --- | --- | --- | --- | 26 (1.48%) | 14 (0.8%) | --- | 119 (6.77%) | 143 (8.13%) | --- |
| Ospina et al. 2020 [28] | Colombia | Cohort in retrospective | Jun15-Jul16 | 5673 | --- | --- | --- | --- | 172 (3%) | --- | --- | 333 (5.87%) | 172 (3.03%) | --- |
| Brasil et al. 2020 [29] | Brazil | Cohort in prospective | Sep15- Feb16 | 511 | --- | --- | --- | --- | --- | 4 (0.78%) | --- | --- | --- | 130/94 (65%) |
| Coutinho et al. 2020 [13] | Brazil | Cohort in prospective | Dec 15- Dec 16 | --- | 16 (3.13%) | 15 (2.93%) | 6 (1.17%) | 6 (1.17%) | 20 (3.91%) | --- | --- | --- | --- | 0/5 |
| Coelho and Crovella 2017 [11] | Brazil | Systematic Review and Meta-analysis | Not specified | --- | Prevalence 2.3% (95% CI 1.0-5.3%) | --- | 42.6% (95% CI 30.8-54.4%) | --- | --- | --- | --- | --- | --- | --- |
| Nithyanantham and Badawi 2019 [12] | Canada | Systematic Review and Meta-analysis | Until Oct17 | --- | Prevalence 3.9% (95% CI 2.4-5.4%) | --- | --- | 21.8% (95% CI 15.2-28.4) | --- | --- | --- | --- | --- | --- |

SGA (Small for Gestacional Age); LBW (Low Birth Weight); RT-PCR (Reverse Transcription Polymerase Chain Reaction); 95% CI (95% Confidence Interval).

RT-PCR for ZIKV in serum or urine were included [23,25–30,33,34,36,39] and ten cohorts included pregnant women with RT-PCR confirmed infection for ZIKV in serum or urine and those with probable infection with IgM measurement with PRNT for ZIKV [18–22,24,35,37,38,40]. In nine studies, only symptomatic pregnant women were included [23,25,27–29,33,34,36,40], in the other 12 studies, both symptomatic and asymptomatic pregnant women were included. Only four studies had a control group with pregnant women without ZIKV infection [19,20,26,34], with different bivariate analysis methods used. The pregnant women included in the studies were infected with ZIKV in different trimesters of pregnancy. Only six studies described the trimester of ZIKV infection [20,24,26,33,34,36]. The analysis of potential confounding factors was performed in a few cohorts such as serology for dengue and Chikungunya [18,34], other serologies [36], maternal comorbidities [26,34] and sociodemographic data [19,20,26,30,34,36].

The microcephaly outcome was analyzed in 16 studies [18–20,22,23,25,26,30,33–40]. These studies used different head circumference growth curves to define microcephaly: Olsen curve [42] (Adhikari et al. 2017) [19], Fenton curve [43] (Mulkey et al. 2019) [40], charts from the World Health Organization (WHO) [44] (Hall et al. 2017 [35]; Honein et al. 2017 [37]; João et al. 2018 [38]; Rice et al. 2018 [22]; Rodriguez-Morales et al. 2018 [23]; Shapiro-Mendoza et al. 2017 [24]) and Intergrowth21st [45] (Hoen et al. 2018 [36]; Rice et al. 2018 [22]; Sousa et al. 2020 [25]; Sanchez Clemente et al. 2020 [26]; Coutinho et al. 2020 [33]). Two studies reported the outcome of microcephaly or CNS abnormalities in the same group, making it not possible to establish the frequency of microcephaly or CNS congenital abnormalities. Only three cohorts differentiate the prevalence of moderate and severe microcephaly (Hoen et al. 2018 [36]; Pomar et al. 2017 [20]; Coutinho et al. 2020 [33]). The prevalence of moderate microcephaly ranged from 1.7 to 4.1% and of severe from 0.3 to 1.6%. Rice et al. 2018 [22] also reported microcephaly developed in the postnatal period. Adams et al. (2016) [18]; Adhikari et al. (2017) [19] and Aspilcueta-Gho et al. (2017) [30] found no conceptus with microcephaly and, in those studies where cases of microcephaly were identified, the frequency ranged from 1.21% [40] to 11.1% [25]. Proportionate and disproportionate microcephaly are described in three studies: Brasil et al. 2016 [34] found a proportion of 50% of proportionate and 50% of disproportionate cases among the four cases of microcephaly, Hoen et al. 2018 [36] found 43.75% of proportionate cases, 28.12% of disproportionate cases among 32 microcephalic neonates, and Sanchez Clemente et al. 2020 [26] described two cases of microcephaly, both disproportionate.

Congenital CNS abnormalities were diagnosed by different imaging exams, using both fetal and cranial ultrasound, cranial computed tomography (CT) or brain magnetic resonance imaging (MRI) [18–25,33,36–38,40]. Adams et al. 2016 [18] and Rodriguez-Morales et al. 2018 [23] did not find any CNS congenital abnormalities in the concepts. In the cohorts that reported CNS congenital abnormalities, the frequency varied widely from 2.38% [36] to 31.7% [40] and the most commonly types described were intracranial calcifications [25,33,36–38,40] and ventriculomegaly [20,25,33,36,38]. The different types of CNS congenital abnormalities described in the studies are discriminated in Table 2, from the most frequent to the least frequent.

Miscarriage was described in 15 studies [18,20–22,24–28,33–37,40]. The frequency at which miscarriage was reported varied from 0.3% [18] to 10.6% [37]. Sanchez Clemente et al. [26] reported no cases of miscarriage. Stillbirth was analyzed as an outcome in nine studies [19,20,25,27,33–37]. Four studies found no cases of stillbirth [19,20,35,37] and in those in which it was reported, the frequency varied from 0.78% [33] to 2.4% [34]. Small for gestational age (SGA) was reported in three cohorts [26,30,34], low birth weigh (LBW) in five studies [19,23,26–28] and prematurity (PMT) in seven studies [19,23,25–28,34]. The frequency varied:

**Table 2. Different types of CNS congenital abnormalities reported in fetuses/neonates of pregnant women with confirmed or probable ZIKV infection.**

| Types of congenital CNS abnormalities described |
| --- |
| Intracranial calcifications [25,33,36–38,40] |
| Ventriculomegaly [20,25,33,36,38] |
| Neural tube defects [30,35] |
| Polymicrogyria [40] |
| Hydrocephalus [35] |
| Lysencephaly [36] |
| Heterotopia [40] |
| Encephalocele [37] |
| Arnold Chiari II malformation [37] |
| White matter injury/bleeding [37] |
| Germinolytic cyst/choroid plexus cyst [37] |
| Anomaly of the corpus callosum [20] |
| Anomalies of the posterior fossa [20] |
| Cerebral hyperechogenicity [20] |
| Abnormal gyration [20] |
| Unspecified CNS abnormalities [18,19,21,33] |

SGA from 1.71% [34] to 9.09% [26], LBW from 1.16% [23] to 9.09% [26] and PMT from 2.32% [23] to 14.5% [34].

We also analyzed the RT-PCR for ZIKV in neonates born from mothers with confirmed/probable ZIKV infection during the pregnancy. This data was available in six studies, and the detection of the virus in serum, urine or cerebrospinal fluid varied from no detection [33] to 65% [29]. Eyes and ear abnormalities were searched throughout the cohorts. Abnormal hearing testing was observed from no cases [33] to 5.9% [25] among the neonates screened. Besides, there was not a specific prevalence rate of eye abnormalities among neonates; some of the findings are described in Table 3.

## Meta-analysis

The meta-analysis of the outcomes studied in the fetuses and neonates of 35,568 pregnant women with confirmed or probable ZIKV infection was performed. CNS abnormality had the highest prevalence ratio of 0.06 (95% CI 0.03–0.09). Intracranial calcifications had a prevalence

**Table 3. Different types of congenital eyes abnormalities reported in neonates of pregnant women with confirmed or probable ZIKV infection.**

| Types of congenital eyes abnormalities described [25,28,33–35] |
| --- |
| Macular lesions |
| Gross macular pigment mottling |
| Macular atrophy and optic nerve hypoplasia |
| Rarefaction of retinal pigment epithelium |
| Pale optic nerve |
| Retinal haemorrhage |
| Strabismus |
| Cataracts |
| Microphthalmia/anophthalmia |
| Coloboma |

ratio of 0.01 (95% CI 0.01–0.02) and ventriculomegaly 0.01 (95% CI 0.01–0.02). Other types of CNS abnormalities did not have enough data to perform a meta-analysis. The prevalence ratio of microcephaly was 0.03 (95% CI 0.02–0.05), fetal loss (miscarriage and stillbirth) was 0.04 (95% CI 0.02–0.06), SGA was 0.04 (95% CI 0.00–0,09), LBW was 0.05 (95% CI 0.03–0.08) and PMT was 0.07 (95% CI 0.04–0.10). The positivity in RT-PCR for ZIKV performed in neonates born to infected mothers during pregnancy was 0.25 (95% CI 0.06–0.44). Given the great variability in the sample size of the different studies analyzed, the heterogeneity found was high, with $I^2 > 90\%$. Heterogeneity was low in the meta-analysis of specific congenital defects: intracranial calcifications, ventriculomegaly, stillbirth and SGA. There was not enough data concerning eyes and ears congenital abnormalities to perform a meta-analysis. Figs 2–10 show the results found in the forest plots generated in the meta-analysis of the different outcomes.

Two recent systematic reviews with published meta-analysis, studying the disorders in fetuses and neonates exposed to ZIKV during pregnancy, found prevalences rates of microcephaly similar to our results. Coelho and Crovella [31] found a prevalence of 2.3% (95% CI 1.0–5.3%) in all studied pregnant women, and Nithiyanantham and Badawi [32] found a prevalence of 3.9% (95% CI 2.4–5.4). We performed the meta-analysis of meta-analysis for microcephaly with the results from other systematic reviews. The final prevalence ratio was 0.03 (95% CI 0.00–0.25). Fig 11 shows the results found in the generated forest plot.

## Discussion

The main objective of this review was to estimate the prevalence rate of disorders in fetuses/ neonates of pregnant women with confirmed or probable ZIKV infection. Analysis of the selected articles has shown a prevalence rate of 3% of congenital microcephaly, 6% of CNS abnormalities, with 1% of intracranial calcifications and of ventriculomegaly, and 4% of fetal loss. We also observed 4% of SGA, 5% of LBW, and 7% of prematurity. RT-PCR for ZIKV was positive in at least one organic fluid in 25% of the neonates.

The pathogenesis of the ZIKV transplacental transmission process, although well accepted in the literature [46], is still poorly understood. ZIKV seems to be able to induce vascular damage and apoptosis in the placental tissue, making the placenta more permeable, facilitating the entry of the virus into syncytiotrophoblast cells. Once in placental tissue, ZIKV can replicate in other cell types such as macrophages and fetal endothelial cells, acting as true deposits of the virus, allowing its spread in fetal blood [47]. Similar to other flaviviruses, cell surface receptors such as Tyro3, Axl and Mert (TAM) appear to play an essential role in the endocytosis process of ZIKV in placental cells [48].

The meta-analysis of meta-analysis, including the results from Coelho and Crovella [31] and Nithiyanantham and Badawi [32], brought a more accurate prevalence ratio of microcephaly. The 3% prevalence rate of congenital microcephaly may seem low, even though this was the first clinical sign that drew attention to congenital Zika syndrome. However, when compared to the prevalence of congenital microcephaly in the pre-ZIKV period, it shows a considerable increase. A study carried out in South America between 2005 and 2014, using data from 107 hospitals in 10 different countries, estimated a prevalence of congenital microcephaly in three cases in 10,000 live births in the general population (0.03%) [49]. Marinho et al. [50] carried out a study based on data from declarations of live births in Brazil, between 2000 and 2015, and found a prevalence of 0.5% in the notification of congenital microcephaly. A systematic review developed by Candelo et al. [51] found an average of 1.8 cases in 10,000 live births (0.02%). Other studies on the prevalence of congenital microcephaly, in the pre-ZIKV period, show higher prevalence rates of moderate forms when compared to severe forms. Silva et al. [52] studied the population data from two Brazilian metropolises and found a prevalence of

**Microcephaly**

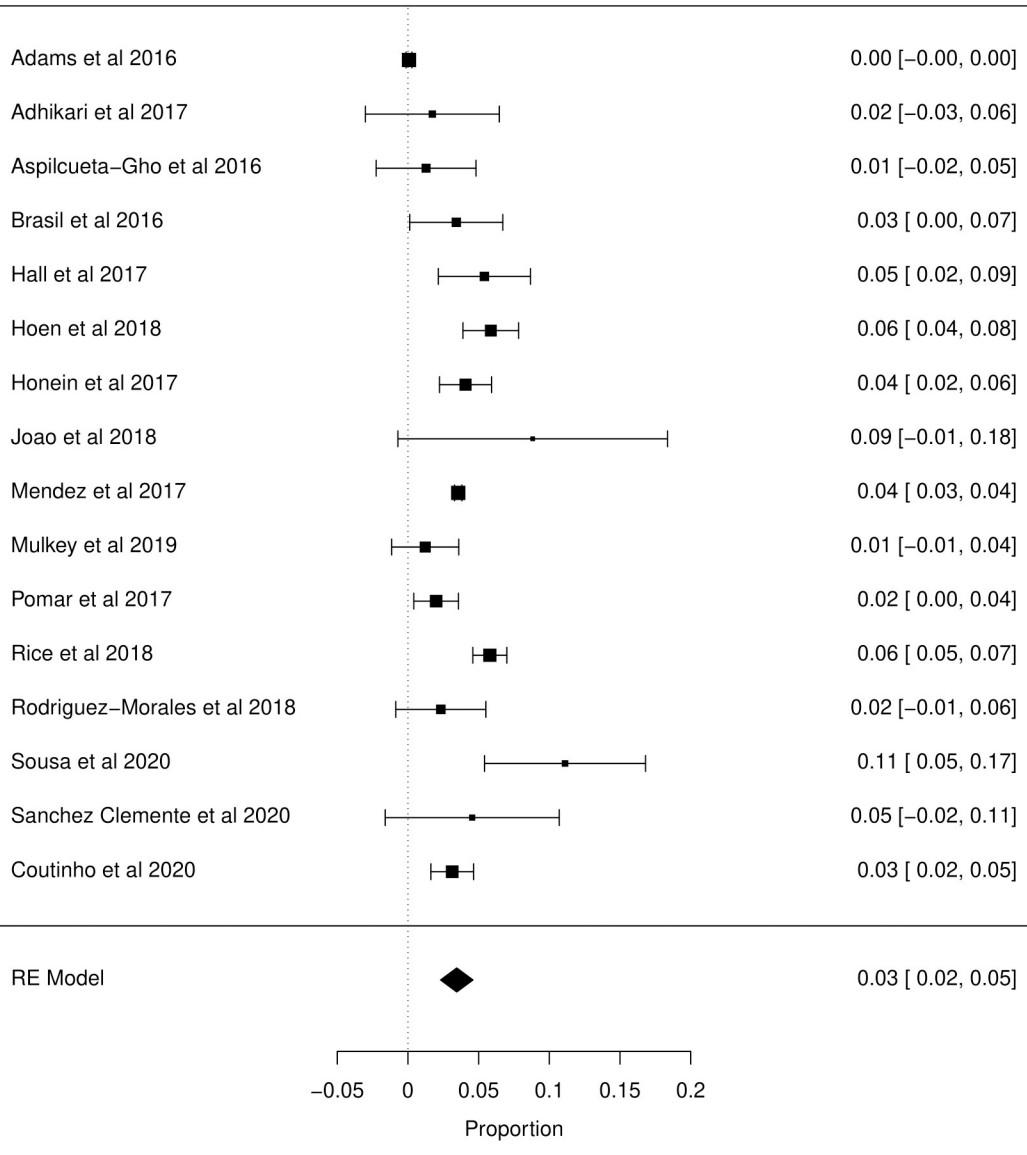

**Fig 2. Meta-analysis of the prevalence rate of microcephaly in neonates born to ZIKV infected mothers.** Random-Effects/Values represent proportions with 95% confidence intervals. Model. $I^2$ (total heterogeneity/total variability): 95.21%. Test for Heterogeneity: Q(df = 15) = 530.1468, p value < 0.0001.

2.5–3.5% in the moderate form and 0.5–0.7% in the severe form. Another Brazilian study estimated a prevalence of 5.6% of moderate forms and 1.5% of severe forms, based on data from neonates admitted to a neonatal intensive care unit in three different cities [53]. Hoyt et al. [54] described a prevalence approximately 3 times higher in the moderate form, in the neonates born in Texas, USA, between 2008 and 2012.

**CNS abnormalities**

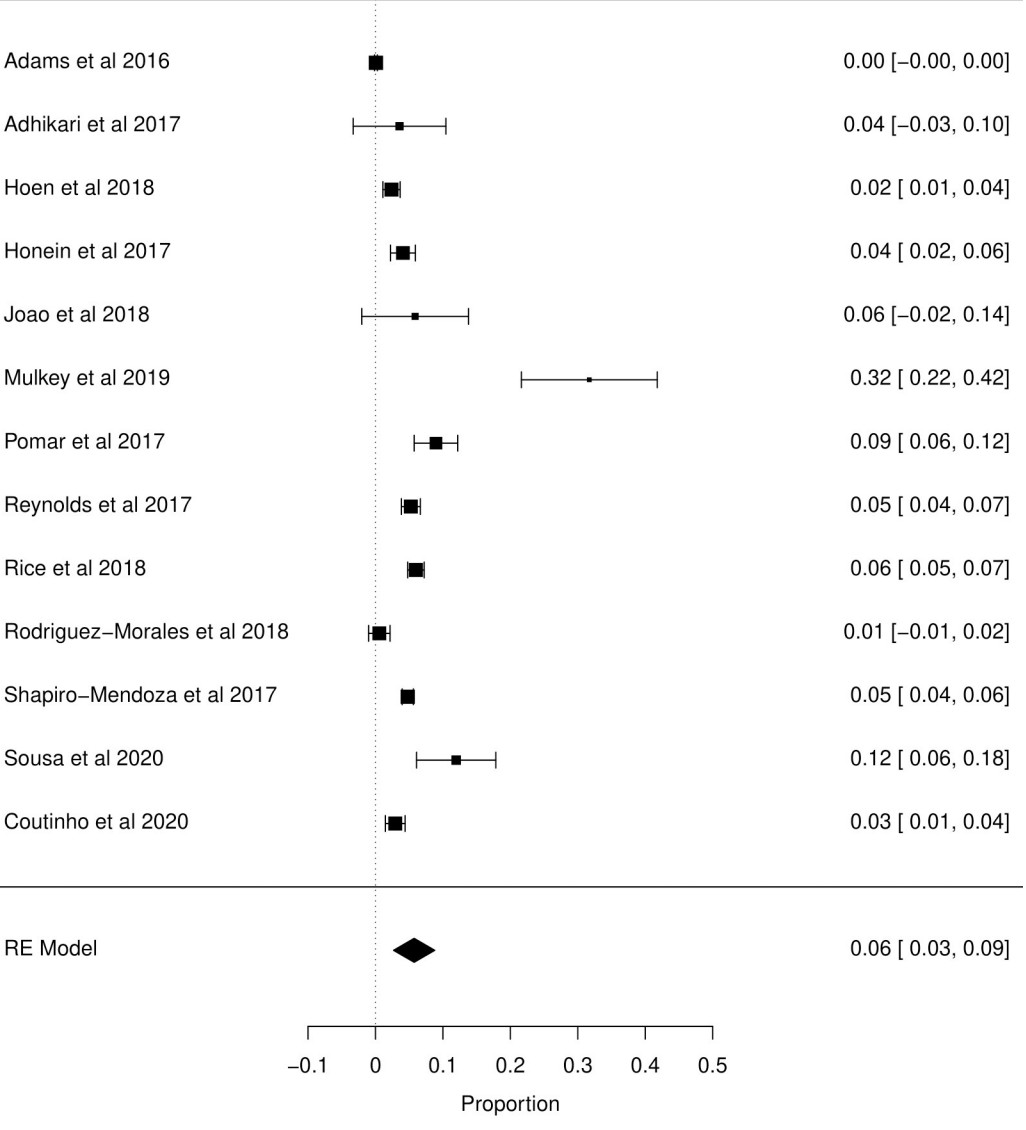

**Fig 3. Meta-analysis of the prevalence rate of CNS abnormalities in neonates born to ZIKV infected mothers.**
Random-Effects/Values represent proportions with 95% confidence intervals. Model. $I^2$ (total heterogeneity/total variability): 98.66%. Test for Heterogeneity: Q(df = 12) = 347.6898, p value < 0.0001.

The variability in the prevalence of congenital microcephaly in the studies in our review (1.21% to 8.82%) could be explained by the different head circumference curves for age and sex used, different definitions of microcephaly utilized at the beginning of Brazil's outbreak, also, the different sample sizes of exposed pregnant women (28 to 19,963 pregnant women). Besides, the non-differentiation of the prevalence of microcephaly in different trimesters in

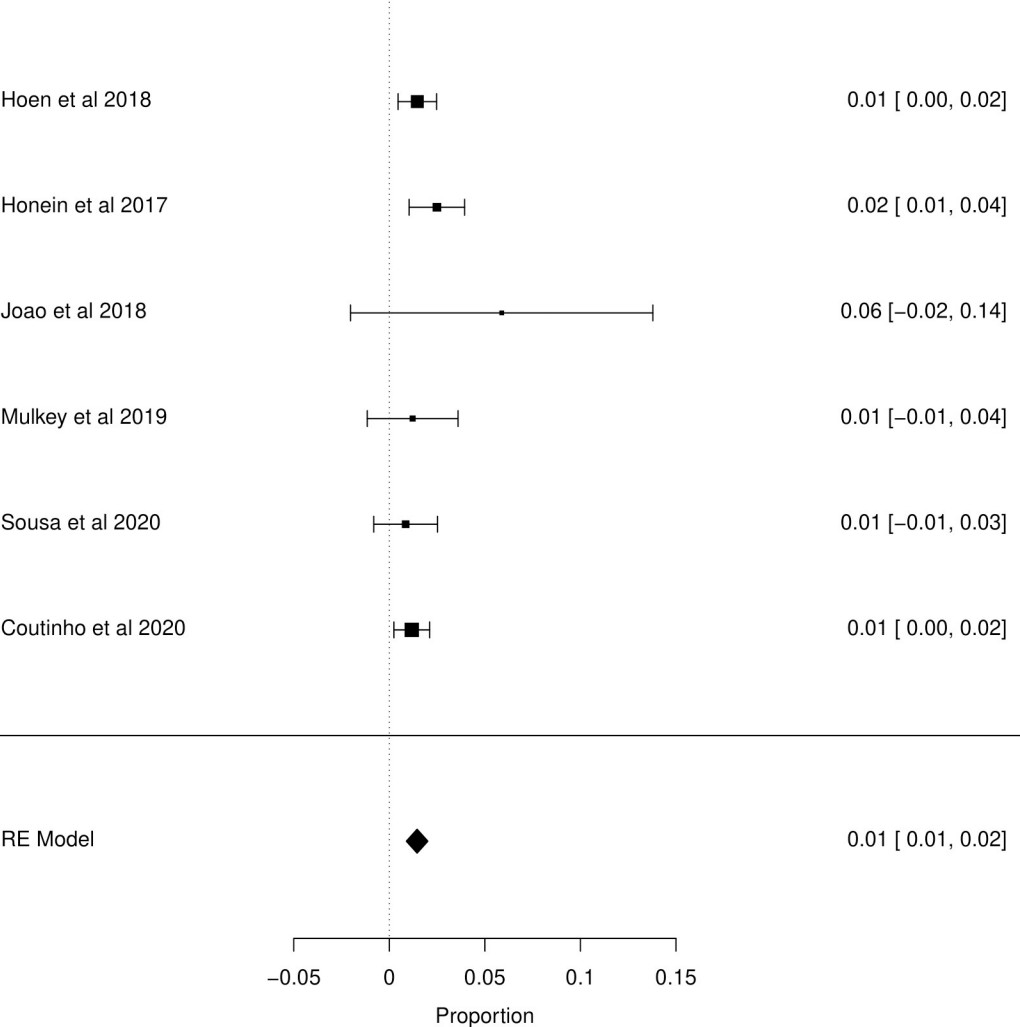

**Intracranial calcifications**

| | |
|---|---|
| Hoen et al 2018 | 0.01 [ 0.00, 0.02] |
| Honein et al 2017 | 0.02 [ 0.01, 0.04] |
| Joao et al 2018 | 0.06 [−0.02, 0.14] |
| Mulkey et al 2019 | 0.01 [−0.01, 0.04] |
| Sousa et al 2020 | 0.01 [−0.01, 0.03] |
| Coutinho et al 2020 | 0.01 [ 0.00, 0.02] |
| RE Model | 0.01 [ 0.01, 0.02] |

**Fig 4. Meta-analysis of the prevalence rate of intracranial calcifications in neonates born to ZIKV infected mothers.** Random-Effects/Values represent proportions with 95% confidence intervals. Model. $I^2$ (total heterogeneity/total variability): 0.51%. Test for Heterogeneity: Q(df = 5) = 4.0328, p value = 0.5447.

which the ZIKV infection had occurred might also have contributed to this difference. Some of the cohorts might have captured more women infected in the first and second trimesters and others with more women infected in the third. Recently, some studies have shown that the prevalence of microcephaly appears to be inversely proportional to the trimester of pregnancy in which the exposure takes place [20,35,55].

**Ventriculomegaly**

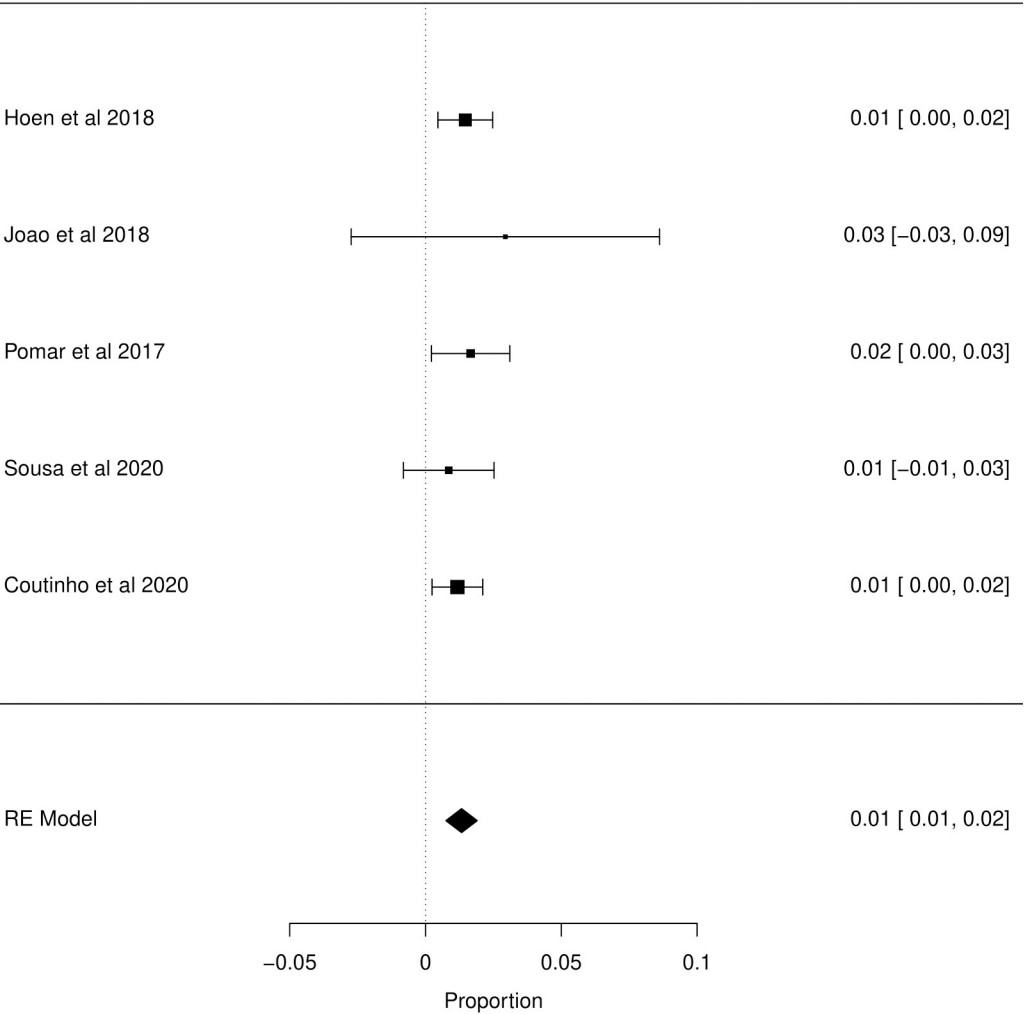

**Fig 5. Meta-analysis of the prevalence rate of ventriculomegaly in neonates born to ZIKV infected mothers.** Random-Effects/Values represent proportions with 95% confidence intervals. Model. $I^2$ (total heterogeneity/total variability): 0.00%. Test for Heterogeneity: Q(df = 4) = 0.9991, p value = 0.9099.

The intrauterine development of the CNS occurs as a complicated and prolonged process, making it susceptible to developmental abnormalities in its different stages. Its prevalence has considerably increased since the clinical use of brain and spinal magnetic resonance imaging. CNS congenital abnormalities represent a heterogeneous group, with hundreds of types of malformations described and different stages of severity. The prevalence of these abnormalities

**Fetal loss**

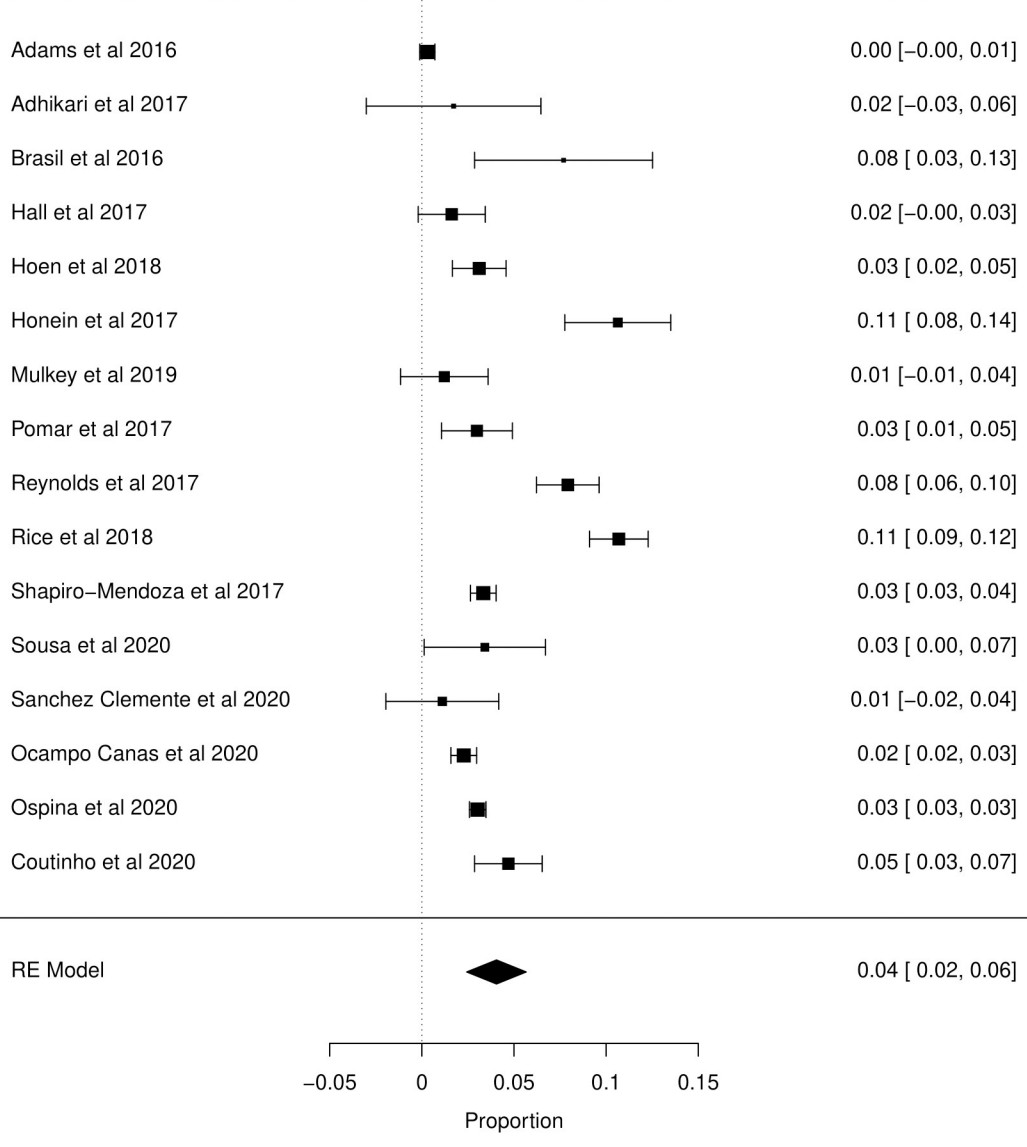

**Fig 6. Meta-analysis of the prevalence rate of fetal loss in ZIKV infected mothers.** Random-Effects/Values represent proportions with 95% confidence intervals. Model. $I^2$ (total heterogeneity/total variability): 97.11%. Test for Heterogeneity: Q(df = 15) = 306.0131, p value < 0.0001.

is estimated at 0.1 to 0.36% in all live births [56–58]. Therefore, the prevalence found of congenital CNS congenital abnormalities in pregnant women exposed to ZIKV (6% - 95% CI 3–9%) is 6 to 12 times higher than that estimated in the general population. Our results showed that, in addition to microcephaly, intracranial calcifications and ventriculomegaly are the CNS congenital abnormalities most commonly found in children exposed to intrauterine ZIKV. In

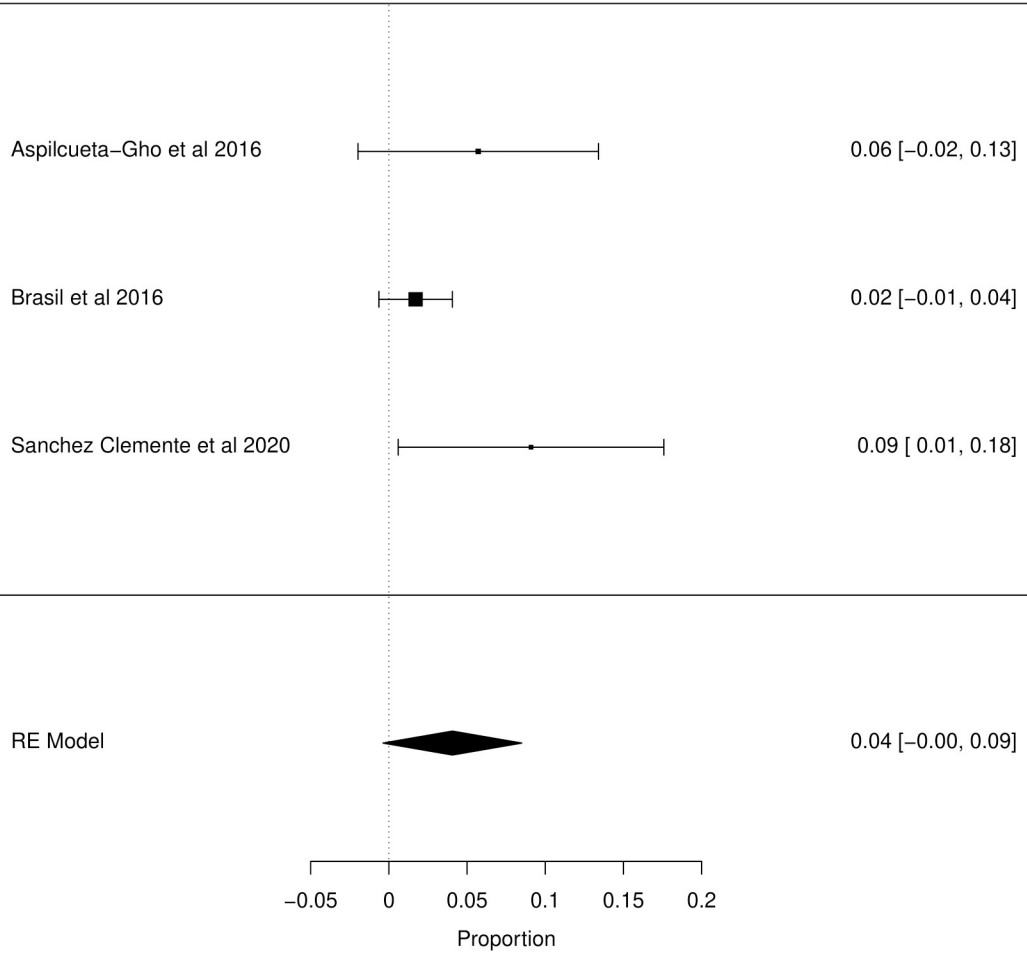

**Small for gestacional age**

**Fig 7. Meta-analysis of the prevalence rate of SGA in neonates born to ZIKV infected mothers.** Random-Effects/
Values represent proportions with 95% confidence intervals. Model. $I^2$ (total heterogeneity/total variability): 43.97%. Test
for Heterogeneity: Q(df = 2) = 3.4191, p value = 0.1810.

a recent systematic review, the prevalence rates of CNS abnormalities were: reduced brain volume (80–81.5%), subcortical calcifications (88.2–93.3%), microcephaly (90–93.3%) and ventriculomegaly (73.3–78.1%). This study also showed that the prevalence of these abnormalities was inversely proportional to the trimester of pregnancy in which the ZIKV infection occurred [55].

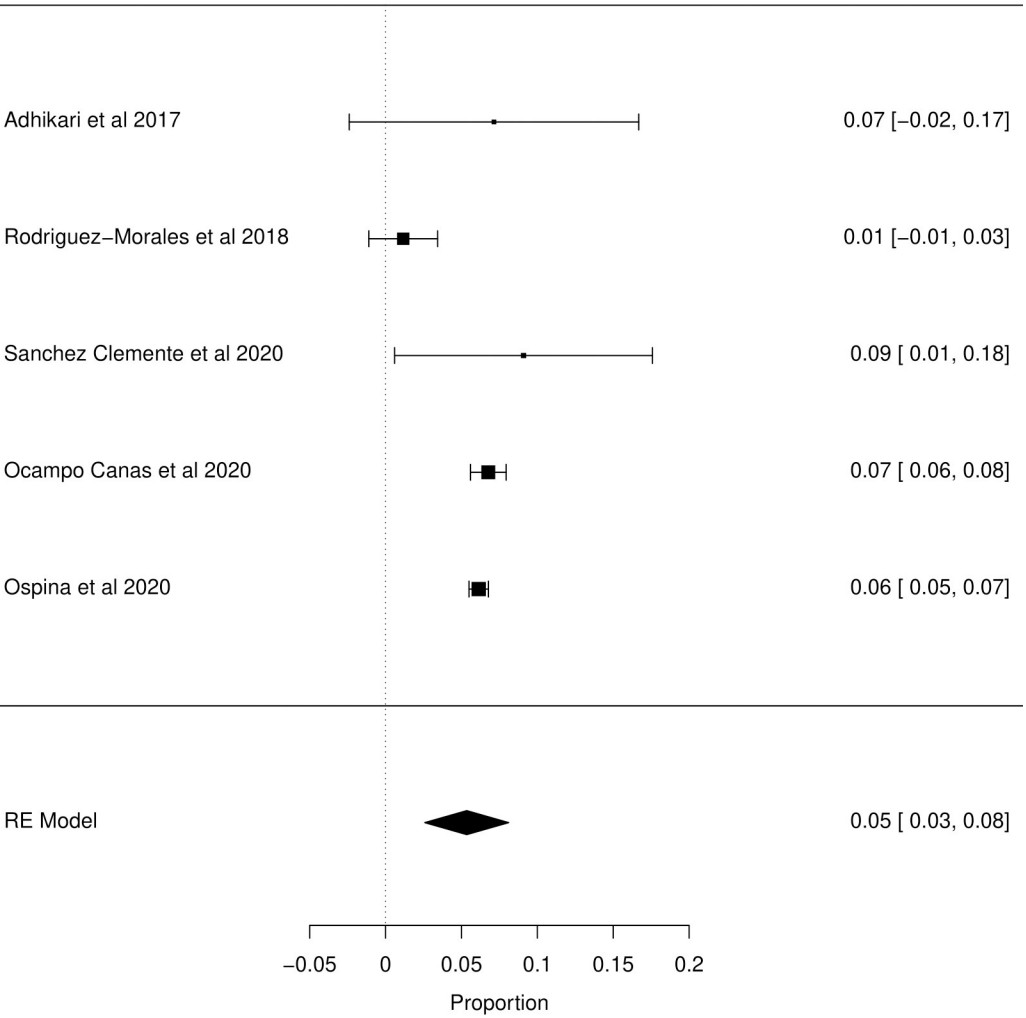

**Fig 8. Meta-analysis of the prevalence rate of LBW in neonates born to ZIKV infected mothers.** Random-Effects/ Values represent proportions with 95% confidence intervals. Model. $I^2$ (total heterogeneity/total variability): 90.04%. Test for Heterogeneity: Q(df = 4) = 19.8976, p value = 0.0005.

Congenital infections have traditionally been included in the differential diagnosis of intracranial calcifications, mainly when it occurs in neonates and infants with other signs and symptoms like microcephaly, low birth weight, hepatosplenomegaly, retinopathy, sensorineural deafness, anaemia, thrombocytopenia and jaundice. Intracranial calcifications are described in congenital infections such as those caused by the cytomegalovirus (CMV),

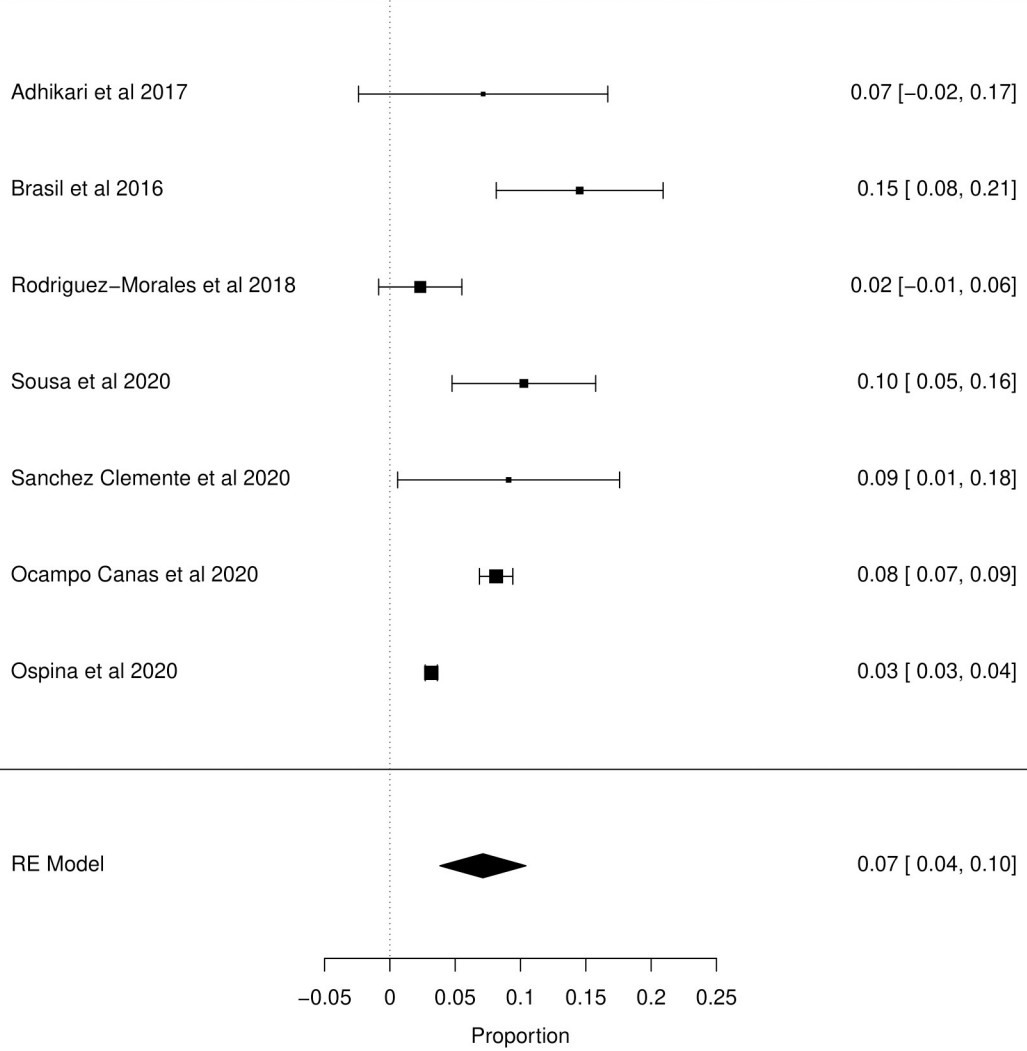

**Fig 9. Meta-analysis of the prevalence rate of prematurity in neonates born to ZIKV infected mothers.** Random-Effects/Values represent proportions with 95% confidence intervals. Model. $I^2$ (total heterogeneity/total variability): 92.39%. Test for Heterogeneity: Q(df = 6) = 70.1163, p value < 0.0001.

toxoplasmosis, Herpes virus, rubella virus and the human immunodeficiency virus (HIV) [59]. Congenital ZIKV infection enters the differential diagnosis of intracranial calcifications, as it is one of its most common findings, with a characteristic pattern of occurrence preferentially between the cortex and the subcortical white matter [60]. It is important to note that the prevalence rate of intracranial calcifications involves children exposed to intrauterine ZIKV but not

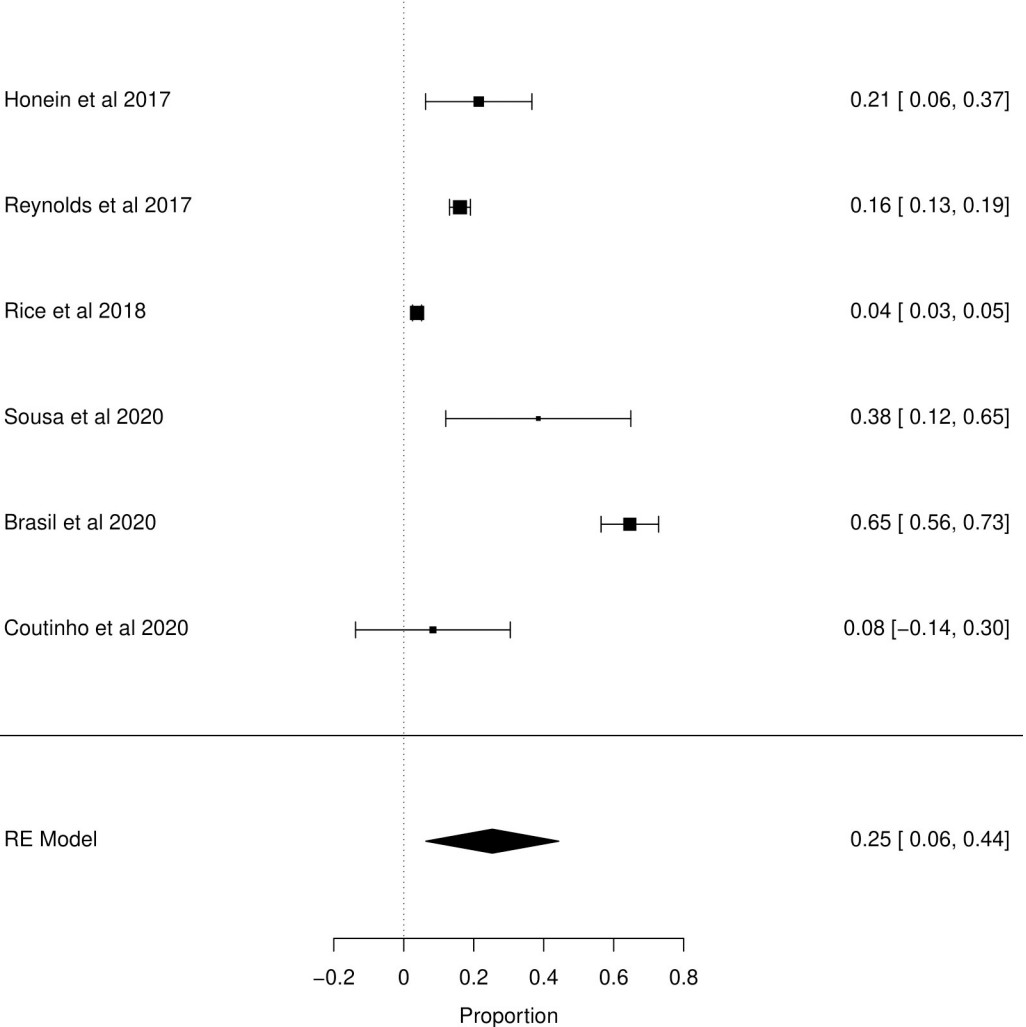

**Fig 10. Meta-analysis of the prevalence rate of positivity in RT-PCR for ZIKV in neonates born to ZIKV infected mothers.** Random-Effects/Values represent proportions with 95% confidence intervals. Model. $I^2$ (total heterogeneity/total variability): 98.89%. Test for Heterogeneity: Q(df = 5) = 257.0044, p value < 0.0001.

adequately diagnosed with congenital Zika syndrome since other congenital infections were not excluded in the majority of the cohorts.

Fetal or congenital ventriculomegaly is the CNS congenital anomaly most commonly detected on prenatal ultrasound, with a prevalence ranging from 1: 250 to 1: 1,600 live births [61]. Ventriculomegaly is a sonographic signal that represents the outcome of different

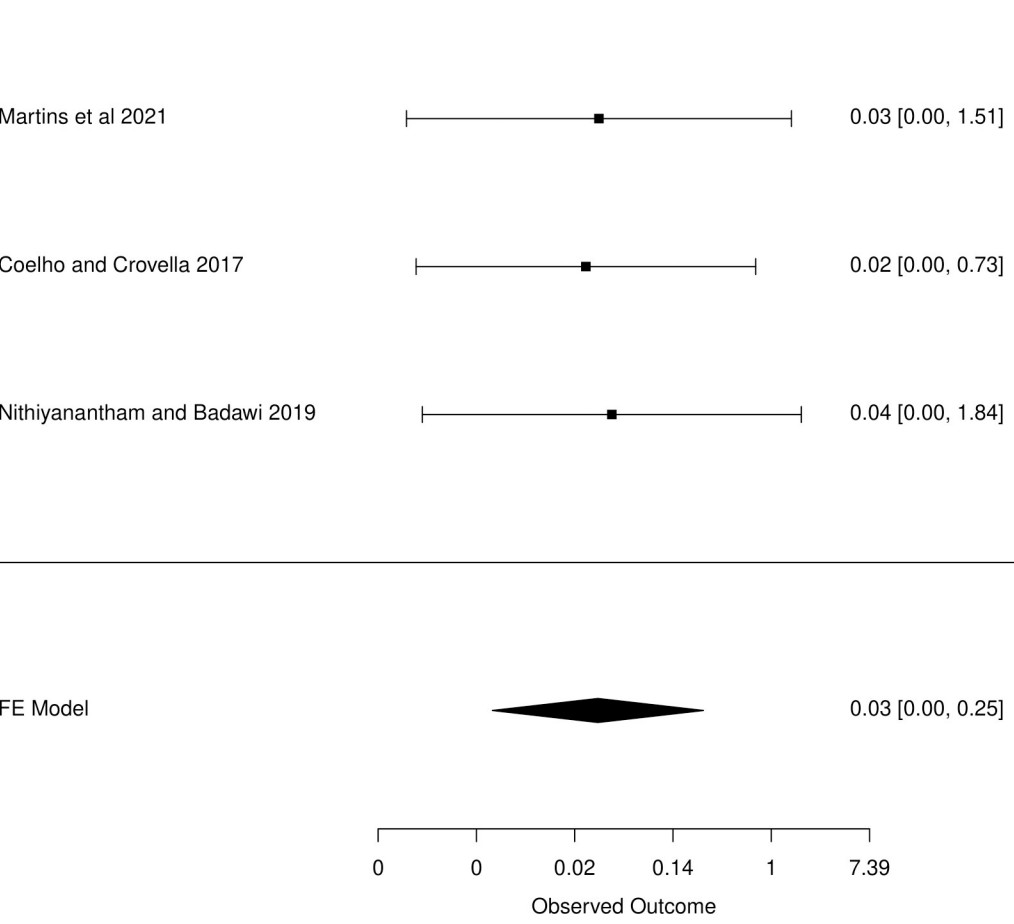

**Microcephaly**

Martins et al 2021 — 0.03 [0.00, 1.51]

Coelho and Crovella 2017 — 0.02 [0.00, 0.73]

Nithiyanantham and Badawi 2019 — 0.04 [0.00, 1.84]

FE Model — 0.03 [0.00, 0.25]

Observed Outcome: 0  0  0.02  0.14  1  7.39

**Fig 11. Meta-analysis of the prevalence rate of microcephaly in neonates born to ZIKV infected mothers.** Random-Effects/Values represent proportions with 95% confidence intervals. Model. $I^2$ (total heterogeneity/total variability): 98.89%. Test for Heterogeneity: Q(df = 5) = 257.0044, p value < 0.0001.

pathological processes with different prognosis. It is usually categorized according to the degree of dilation between mild (10–12 mm), moderate (13–15 mm) and severe (> 15 mm) [62] ocurrences. The aetiology, the presence of other associated abnormalities, the degree of severity and the progression of ventriculomegaly are the main determinants in its impact on the neurodevelopment of fetuses and neonates [61]. The etiologies of congenital

ventriculomegaly are diverse and can be divided into causes that lead to loss of brain tissue, causes that obstruct the ventricular system and those that lead to excessive production of cerebrospinal fluid (CSF) [63]. Approximately 5% of mild and moderate ventriculomegaly cases result from congenital fetal infections, including CMV, toxoplasmosis and ZIKV. Isolated cases of ventriculomegaly associated with other viruses are described (mumps, enterovirus 71, parainfluenza virus type 3, parvovirus B19 and the lymphocytic choriomeningitis virus) [62]. In congenital infections, ventriculomegaly may or may not be associated with other congenital malformations, including others in the CNS itself, or occur in isolation. Ventriculomegaly secondary to congenital infections is the result of a process of cerebral atrophy or inflammatory arachnoid granulations that lead to obstruction of the ventricular system [63]. Ventriculomegaly is a widespread finding in children with symptomatic congenital Zika syndrome [55]. The prevalence range of ventriculomegaly, regardless of the aetiology, in the general population, is around 0.06 to 2% of fetuses [61].

Miscarriage is defined as the fetal loss that occurs before 20 weeks of pregnancy in 15 to 20% of pregnancies. While up to 50% of miscarriages occur secondary to embryonic chromosomal imbalances, its aetiology in fetuses with a normal karyotype is still not well understood. Many risk factors were identified as: maternal age, medication use, maternal overweight or malnutrition, alcohol, smoking, in addition to genetic factors [64]. Maternal infections during pregnancy can also represent a risk factor. Infection with parvovirus B19 during pregnancy has a cumulative incidence of miscarriage of approximately 8%, and the risk is 5.6 times higher in those with infection in the first trimester of pregnancy [65]. Untreated syphilis during pregnancy leads to a 21% increase in the risk of miscarriage and stillbirth [66]. ZIKV may have a role as a risk factor for fetal loss as it occurs with other congenital infections.

Preterm birth remains a significant public health priority worldwide. Preterm birth may be considered as an adverse pregnancy outcome (where a fetus is unable to fulfil in utero growth potential) or a preferred outcome (where a miscarriage or nonviable prematurity has been successfully avoided) [67]. Some recent studies have published global and regional estimates of preterm birth incidence in the general population, which varies between 9.1% and 11.1% [68–70]. Several risk factors for preterm birth are described, among them, the vertical infection transmission [71]. Maternal and fetal infections also seem to be risk factors for SGA [72] and LBW [73,74], and their incidence was reported in both SGA [75] and LBW [70].

Laboratory confirmation of ZIKV infection is challenging due to its short window of viremia and virus, enabling RT-PCR detection. Indeed, the duration of ZIKV viremia and viruria in vertically infected children is unknown, and it is unclear whether fetuses infected early during intrauterine life have detectable virus at birth, as the duration of viral shedding from intrauterine infection has not been described yet. Also, it is unclear whether viral presence in blood, urine or cerebrospinal fluid in vertically infected children is constant or intermittent, as in those children infected early during pregnancy the viral infection could be gone by the time of birth and only the sequelae of infection are present [29]. These factors might explain the wide variation in the prevalence of positive exams in different studies, ranging from no cases [40] to 65% [29]. Rice et al. [22] performed an analysis in the American database of children exposed to ZIKV during their fetal period, as Brasil et al. [29] clinically followed the children and performed the exams during the ZIKV outbreak in Brazil, one could argue that a large number of positives could result from postnatal exposure to the virus. Lastly, comparing different study designs make it difficult to compare the results.

Some strengths of our review should be highlighted: broad search strategy, including grey literature, reducing the odds of publication bias; and the inclusion of articles published in Portuguese, Spanish and English, the languages are spoken in almost all of the Americas, where most of the reports of ZIKV infections in pregnant women occurred. Our systematic review

has some limitations, such as publication bias and heterogeneity. We tried to minimize publication bias with extensive bibliographic research, including conference proceedings, theses and dissertations and grey literature, all in more than one scientific database. Heterogeneity is expected in systematic reviews and meta-analyses of observational studies and will always be an inherent limitation. Different criteria might explain the heterogeneity in our study for outcomes definition in the different cohorts analyzed, in addition to the different sample sizes and their different places of occurrence, consequently with different health structures. Different criteria, especially concerning the studied outcomes, may have contributed to the heterogeneity found.

Our study analyzed cohorts describing outcomes diagnosed in the prenatal period or shortly after birth, except a single study by Rice et al. [22] who also analyzed longer-term outcomes. It might play a limiting factor in our review, as recent studies describing clinical follow-up of children exposed to intrauterine ZIKV during the first two years of the life, even in those children born without congenital anomalies, have shown that they can develop postnatal microcephaly or impaired neurodevelopment [76–79].

## Conclusions

Our study analyzed the prevalence of disorders in fetuses and neonates of pregnant women with probable or confirmed ZIKV infection, such as microcephaly, CNS congenital abnormalities, intracranial calcifications, ventriculomegaly, fetal loss, small for gestational age, low birth weight and prematurity. Our results estimated the impact on children exposed to ZIKV infection during pregnancy and highlighted the high prevalence of microcephaly, CNS congenital abnormalities and fetal loss. The importance of maintaining studies in the area should be emphasized, especially in those whose main objective is postnatal monitoring, since it is known that more than a congenital syndrome with classic signs and symptoms we probably face a spectrum disease, with children being born normocephalic and progressing to postnatal microcephaly or abnormalities of neuropsychomotor development. The continuous knowledge of its magnitude is essential for the development of health measures and programs, in addition to promoting disease prevention, especially in research for the development of the ZIKV vaccine.

## Supporting information

**S1 Checklist. PRISMA 2009 checklist.**
(DOC)

**S1 Table. Full eletronic search strategy.**
(DOCX)

**S2 Table. Critical appraisal checklist for studies reporting prevalence data.**
(DOCX)

**S3 Table. Quality of 21 included cohort studies.**
(DOCX)

## Author Contributions

**Conceptualization:** Marlos Melo Martins, Antonio José Ledo Alves da Cunha, Jaqueline Rodrigues Robaina, Roberto de Andrade Medronho.

**Data curation:** Jaqueline Rodrigues Robaina.

**Formal analysis:** Marlos Melo Martins, Antonio José Ledo Alves da Cunha, Roberto de Andrade Medronho.

**Investigation:** Marlos Melo Martins, Jaqueline Rodrigues Robaina.

**Methodology:** Marlos Melo Martins, Antonio José Ledo Alves da Cunha, Jaqueline Rodrigues Robaina, Carlos Eduardo Raymundo, Arnaldo Prata Barbosa, Roberto de Andrade Medronho.

**Project administration:** Marlos Melo Martins, Jaqueline Rodrigues Robaina, Roberto de Andrade Medronho.

**Software:** Carlos Eduardo Raymundo.

**Supervision:** Antonio José Ledo Alves da Cunha, Arnaldo Prata Barbosa, Roberto de Andrade Medronho.

**Validation:** Carlos Eduardo Raymundo.

**Visualization:** Jaqueline Rodrigues Robaina.

**Writing – original draft:** Marlos Melo Martins.

**Writing – review & editing:** Antonio José Ledo Alves da Cunha, Arnaldo Prata Barbosa, Roberto de Andrade Medronho.

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
