## [Decision Letter · Decision Letter 0]

12 Oct 2020

PONE-D-20-27361

Fetal, neonatal, and infant outcomes associated with maternal Zika virus infection during pregnancy: a systematic review and meta-analysis.

PLOS ONE

Dear Dr. Martins,

Thank you for submitting your manuscript to PLOS ONE. After careful consideration, we feel that it has merit but does not fully meet PLOS ONE’s publication criteria as it currently stands. Therefore, we invite you to submit a revised version of the manuscript that addresses the points raised during the review process.

Two experts in the field handled your manuscript, and we are very thankful for their time and efforts. Although interest was found in your review and meta-analysis, numerous comments and concerns arose. Notably, the references need to be updated and the novelty is in question. There are helpful suggestions from the reviewers to increase the novelty and readability of this manuscript. Please address ALL of the reviewers' comments in your revised manuscript.

We look forward to receiving your revised manuscript.

Kind regards,

Frank T. Spradley

Academic Editor

PLOS ONE

2. Please confirm that you have included all items recommended in the PRISMA checklist including an assessment of publication bias using graphical methods (e.g. Funnel plot) and statistical methods (e.g. Egger’s test) as appropriate.

Reviewers' comments:

Reviewer's Responses to Questions

**Comments to the Author**

1. Is the manuscript technically sound, and do the data support the conclusions?

Reviewer #1: Partly

Reviewer #2: Yes

2. Has the statistical analysis been performed appropriately and rigorously? 

Reviewer #1: I Don't Know

Reviewer #2: Yes

3. Have the authors made all data underlying the findings in their manuscript fully available?

Reviewer #1: Yes

Reviewer #2: Yes

4. Is the manuscript presented in an intelligible fashion and written in standard English?

Reviewer #1: No

Reviewer #2: Yes

5. Review Comments to the Author

Reviewer #1: Thank you for asking me to review this manuscript.

Although this is a useful exercise and has the potential to make an interesting paper, there are several issues that need addressing.

Firstly, the review is already very out of date as the search was not updated after December 2019. There are several publications that were published subsequently that should be included in the review and meta-analysis, including the following paper as well as several others: Zika virus infection in pregnancy and adverse fetal outcomes in São Paulo State, Brazil: a prospective cohort study

NS Clemente, EB Brickley, ES Paixão, MF De Almeida, RE Gazeta, ...

Scientific reports 10 (1), 1-10

The review should, in my opinion also include other important perinatal outcomes including small for gestational age or low birthweight and prematurity. Particularly as the authors said, there already several SRs and meta-analyses looking at microcephaly and other CNS outcomes, so this review needs to bring something novel.

The other SRs and meta-analyses should be incorporated into this review, in my opinion, using appropriate techniques to do meta-analysis of meta-analyses. This would provide a more accurate estimate of the outcomes. Otherwise, it is difficult to interpret these findings.

The discussion/conclusion in particular is really quite poor, there many grammatical errors, difficult to read, rambling in nature and not really discussing the findings of the SR.

Below are a few more specific points for the authors to conider:

Lines 62-65: long sentence and you lose the point. Suggest to revise. Did increased surveillance mean that other states started reporting it? Or other states were reporting this at the same time as well and the surveillance systems picked up on this? For example, I think there were cases being of microcephaly being reported in Paraiba at the same time as the Pernambuco ones

Line 87: The prevalence of fetal and neonatal disorders in pregnant women with proven ZIKV infection – This sentence is not correct as fetal and neonatal disorders by definition do not occur in pregnant women. Also, it would be good to be more specific rather than just saying ‘fetal and neonatal disorders’ for example perinatal outcomes and CNS findings at birth?

Lines 89-92: Again, this could be re-phrased to be more engaging and specific

Lines 92-94: Maybe add some more specific examples, otherwise the sentence sounds very generic

100-105: Maybe present as a diagram or with cleaner text/numbering?

111: It would be good to know which included studies were grey literature?

130: ‘Seccional’ – what do you mean by this? Is it cross-sectional studies?

If this is the case – the abstract implies only cohort studies were included whereas this implies it was cross sectional and cohort

148: Abortion implies termination of pregnancy – do you mean this? Or do you mean early miscarriage?

Lines 157-159: Repetition of lines 140-142

168: magazine – replace with journal

170: number of exposed and unexposed what? Please specify

170-171- Repetition of exposed and unexposed. Also, how did you measure confounding and interaction?

176-178: Please check grammar

Lines 190-2: Please explain this for the reader- is this a good agreement? What are the parameters for this test?

196: The reasons for excluding the other 36 articles were – where did this 36 come from?

And how did you get from 1233 after removal of duplicates to 51 articles?

198: Were the SRs you excluded relevant to your study question? This should be clarified

204-209: Not sure if it is worth listing the studies here as the information is in the table below

Table 1 – typo – observational and accent on Colombia. It would be best to use the the correct epidemiological terminology for these i.e) cohort in prospective. If retrospective – are they case controls? Were they looking at the adverse outcome first and then obtaining exposure variables retrospectively? Or are the cross sectional? You mention you included cross-sectional studies but then none are cross sectional study design in the table.

Line 256: prospects – typo – but please see above

268: 21st not 21th

269-271: Please make your point a bit clearer – you mean those without microcephaly but with CNS abnormalities or ocular/other abnormalities?

272: It would be good to know how many studies looked at proportionate and disproportionate microcephaly as well

280: Usually known as cranial ultrasound rather than transfontanelle. And please write CT in brackets after computed tomography

281: magnetic resonance imaging (MRI)

I would be tempted just to reference the studies rather than list the authors every time you talk about the findings for ease of reading, unless you are talking about one specific study, like in line 284

287: which – typo

291: described x2 – repetitive – please revise

294: newborns – typo

Table 2: heart attack????

310: conceptuses – this is not correct – please revise

312: “ showed that the presence of at least one CNS abnormality had the highest prevalence ratio 0.06 (95% 313 CI 0.02-0.09).” Please revise the structure of this sentence

315: Is prevalence ratio the best way of expressing this?

Stillbirth – does that mean no studies ad any women who had stillbirths?

IS it not worth joining fetal loss/stillbirth/early or late miscarriage as one category?

368-386 – I don’t really understand why this bit of the discussion. You can’t really compare European data from one period to Brazilian data from another period. Also, this is surely a systematic review looking at the prevalence of microcephaly so why is this talking about other studies not in this SR. The discussion should start off summarising the salient points of the study. This seems like a long ramble into the pathophysiology of transplacental transmission which, although interesting, is missing the point of your SR a little.

388: “can be explained mainly” may be best to say: “could be explained by”

390: “in addition to the fact that population samples of exposed pregnant women of very different sizes (28 to 19,963 pregnant women)” – please revise structure/grammar

393: “Besides, the non-differentiation of the prevalence of microcephaly for exposure to ZIKV in different trimesters may also have contributed to this difference,” Do you mean some studies might have captured more women infected in the first trimester and others more women infected in the third? Please explain further in the text

396-398: Please revise grammar and structure

400: It would only increase prevalence if these were in utero antenatal surgical procedures which I don’t think they are?

401-403: Please revise

404: Ref 48 is from a textbook so likely out of date. Also – need to compare prevalence in the same population otherwise can’t draw comparisons.

412-413: Please revise

414-415: Especially in newborns/infants? Surely this is the only category you think about congenital infections in?

417: “intracranial calcification occurs in 30 to 90% of cases of congenital cytomegalovirus (CMV) infection and 50 to 80% of cases of congenital toxoplasmosis” This seems very high – are you sure this is correct? Which country is this data from?

423-425: Please revise – unable to follow

426: Adolescence?? With congenital ZIKV?

463-465: It seem strange to quote one paper and then say it is in agreement with your SR? Surely this paper should be included in the SR?

467: Disorders in concepts – please revise, concepts not correct term here

466: typo – analyses. Also – if meta-analyses already exist, these should have been incorporated into this meta-analysis using appropriate techniques to make it a more all-encompassing review

471-481: please review grammar and structure

481: Different outcomes or different definitions or defining criteria?

485: repetition, study, studies – please revise

487-490: Please revise grammar

495: “disorders in fetuses/newborns” – again, be more specific

496-498 and conclusion as a whole: Please revise grammar. You have not summarised here what your findings were. These are quite generic comments. It would be good to have closing comments based on your findings

Reviewer #2: The authors performed in December 2019 a well planned and executed meta-analysis of pregnancy and neonatal outcomes associated with maternal Zika virus infection during pregnancy. Among 15 cohorts included in the systematic review, they were able to analyze miscarriage , stillbirths, microcephaly, CNS abnormalities, intracranial calcifications and ventriculomegaly. Due to the heterogeneity of the data, not all cohorts contributed to all endpoint meta-analysis. The manuscript is technically sound, well written and adequately approach the findings in a comprehensive manner in the discussion. The overall findings had already been known in the literature. A missing point that might bring more light into the issue would to characterize what is now generally accepted as typical congenital ZIKV syndrome findings . Among those carrying CNS abnormalities/microcephaly what would be the prevalence rate of severe typical CNS involvement? these data might be available in some of the selected publications. If so, it could be analyzed in the manuscript. Also, it would be helpful to know if other secondary outcomes such as eye, ear abnormalities have been searched for.

6. PLOS authors have the option to publish the peer review history of their article (what does this mean?). If published, this will include your full peer review and any attached files.

Reviewer #1: No

Reviewer #2: **Yes: **Marisa Marcia Mussi-Pinhata

---

## [Author Response · Author response to Decision Letter 0]

14 Dec 2020

Frank T. Spradley

Academic Editor

PLOS ONE

Dear Mr Spradley

We would like to express our full appreciation for allowing us to revise our manuscript entitled " Fetal, neonatal, and infant outcomes associated with maternal Zika virus infection during pregnancy: a systematic review and meta-analysis ". We believe that the reviewers made insightful suggestions that immensely helped to address critical unresolved issues to improve the manuscript ultimately. All the changes in the paper are highlighted in "red" in the revised version (Revised Manuscript Track Changes). We made modifications in the document according to the reviewers' comments. We feel that the manuscript is now greatly improved.

In this letter, I respond to each point raised by the reviewers, with the location in the manuscript.

I remain at your disposal to clarify any pending point.

Yours sincerely,

Marlos Melo Martins

Universidade Federal do Rio de Janeiro

Rua das Laranjeiras 180, Rio de Janeiro, RJ, Brazil

Zip Code: 22240-000

 

Reviewer #1: 

Thank you very much for each comment. Your insightful suggestions immensely helped to address critical unresolved issues to improve the manuscript ultimately.

1- Although this is a useful exercise and has the potential to make an interesting paper, there are several issues that need addressing. Firstly, the review is already very out of date as the search was not updated after December 2019. There are several publications that were published subsequently that should be included in the review and meta-analysis, including the following paper as well as several others: Zika virus infection in pregnancy and adverse fetal outcomes in São Paulo State, Brazil: a prospective cohort study NS Clemente, EB Brickley, ES Paixão, MF De Almeida, RE Gazeta, ... Scientific reports 10 (1), 1-10

Answer: We updated our review up to November 2020, with the inclusion of five new cohorts published in 2020, including Sanchez Clemente at al. 2020 as requested.

Location: Page 11, Table 1.

2- The review should, in my opinion also include other important perinatal outcomes including small for gestational age or low birthweight and prematurity. Particularly as the authors said, there already several SRs and meta-analyses looking at microcephaly and other CNS outcomes, so this review needs to bring something novel.

Answer: We included all the outcomes suggested and added the positivity of RT-PCR in children exposed. All the meta-analysis were performed.

Location: Lines 164-167; lines 188 and 189; lines 305-308; lines 309-312; lines 361-376; lines 507-530. Table 1. Figs 7-10.

3- The other SRs and meta-analyses should be incorporated into this review, in my opinion, using appropriate techniques to do meta-analysis of meta-analyses. This would provide a more accurate estimate of the outcomes. Otherwise, it is difficult to interpret these findings.

Answer: We performed, as suggested, the meta-analysis of the meta-analysis.

Location: Lines 378-385, Fig 11. Lines 410-412.

4- The discussion/conclusion in particular is really quite poor, there many grammatical errors, difficult to read, rambling in nature and not really discussing the findings of the SR.

Answer: Suggestion accepted. We modified the discussion and the conclusions, correcting the grammatical errors, trying to make it easier to read and discussing the findings of the SR.

Location: Lines 394-569.

5- Few more specific points for the authors to consider:

Lines 62-65: long sentence and you lose the point. Suggest to revise. Did increased surveillance mean that other states started reporting it? Or other states were reporting this at the same time as well and the surveillance systems picked up on this? For example, I think there were cases being of microcephaly being reported in Paraiba at the same time as the Pernambuco ones

Answer: Suggestion accepted. The sentence was revised and we changed the information as suggested.

Location: Lines 65-67.

Line 87: The prevalence of fetal and neonatal disorders in pregnant women with proven ZIKV infection – This sentence is not correct as fetal and neonatal disorders by definition do not occur in pregnant women. Also, it would be good to be more specific rather than just saying ‘fetal and neonatal disorders’ for example perinatal outcomes and CNS findings at birth?

Answer: Thank you for the comment. We corrected the sentence about prevalence and accepted the suggestion to change to “perinatal outcomes”.

Location: Line 89.

Lines 89-92: Again, this could be re-phrased to be more engaging and specific

Answer: Suggestion accepted, lines re-phrased.

Location: Lines 92-98.

Lines 92-94: Maybe add some more specific examples, otherwise the sentence sounds very generic

Answer: Suggestion accepted, lines re-phrased.

Location: Lines 98-101.

100-105: Maybe present as a diagram or with cleaner text/numbering?

Answer: Presented with numbering.

Location: Lines 107-113.

111: It would be good to know which included studies were grey literature?

Answer: None selected studies were from grey literature. Sentence added.

Location: Lines 217-218.

130: ‘Seccional’ – what do you mean by this? Is it cross-sectional studies?

If this is the case – the abstract implies only cohort studies were included whereas this implies it was cross sectional and cohort

Answer: Thank you for the comment. We corrected the sentence; only cohorts studies were searched.

Location: Lines 142-146.

11- 148: Abortion implies termination of pregnancy – do you mean this? Or do you mean early miscarriage?

Answer: We corrected the sentence. We extracted data from cases of miscarriage (<20 weeks of gestational age) and stillbirth (= or > 20 weeks of gestational age).

Location: Lines 160-161.

Lines 157-159: Repetition of lines 140-142

Answer: Phrase deleted.

168: magazine – replace with journal

Answer: Word replaced.

Location: Line 180.

170: number of exposed and unexposed what? Please specify

Answer: number of exposed and unexposed pregnant women to ZIKV infection.

Location: Lines 181-182.

170-171- Repetition of exposed and unexposed. Also, how did you measure confounding and interaction?

Answer: In those studies which compared pregnant women with and without ZIKV infection, we extracted all other characteristics that could represent possible confounding and interaction factors.

Location: Lines 182-185.

176-178: Please check grammar

Answer: re-phrased

Location: Lines 189-191.

Lines 190-2: Please explain this for the reader- is this a good agreement? What are the parameters for this test?

Answer: Suggestion accepted. The parameters and its explanation were added.

Location: Lines 210-213.

196: The reasons for excluding the other 36 articles were – where did this 36 come from?

And how did you get from 1233 after removal of duplicates to 51 articles?

Answer: The 36 articles came from different databases utilized. We got from 1340 to 58 articles (new numbers since we expanded the period of the study), after the title and abstract screening, performed independently by two reviewers and agreement between us.

Location: Lines 206-207. Lines 218-222.

198: Were the SRs you excluded relevant to your study question? This should be clarified

Answer: we incorporated the SRs in our study, performing a meta-analysis of the meta-analysis as suggested previously.

Location: Page 11, Table 1. Lines 237 -238; lines 378-385. Page 20, Fig 11.

204-209: Not sure if it is worth listing the studies here as the information is in the table below. Table 1 – typo – observational and accent on Colombia. It would be best to use the the correct epidemiological terminology for these i.e) cohort in prospective. If retrospective – are they case controls? Were they looking at the adverse outcome first and then obtaining exposure variables retrospectively? Or are the cross sectional? You mention you included cross-sectional studies but then none are cross sectional study design in the table.

Answer: As we searched only for cohorts, we classified than in prospective and retrospective studies, depending on the moment of collecting the data. Case controls studies were excluded, since our main goal was to estimate the frequency of the outcomes, and it would not be possible with case-control studies. All the studies selected, prospectives and retrospectives, initiated searching the pregnant women with ZIKV infection for posterior analysis of the outcomes. The terms were corrected as requested (cohort in prospective and cohort in retrospective).

Location: Pages 11-12. Table 1.

Line 256: prospects – typo – but please see above

Answer: We decided to delete the sentence since this information is already available in Table 1, Pages 11-12.

268: 21st not 21th

Answer: corrected.

Location: Line 269.

269-271: Please make your point a bit clearer – you mean those without microcephaly but with CNS abnormalities or ocular/other abnormalities?

Answer: Two studies report the outcome of microcephaly or CNS abnormalities in the same group, making it not possible to establish the frequency of microcephaly or CNS congenital abnormalities. The sentence was re-phrased. 

Location: Lines 270-273.

272: It would be good to know how many studies looked at proportionate and disproportionate microcephaly as well

Answer: The information was added to the manuscript.

Location: Lines 279-284.

280: Usually known as cranial ultrasound rather than transfontanelle. And please write CT in brackets after computed tomography

Answer: corrected.

Location: Line 286.

281: magnetic resonance imaging (MRI). 

Answer: corrected as suggested.

Location: Line 287. 

I would be tempted just to reference the studies rather than list the authors every time you talk about the findings for ease of reading, unless you are talking about one specific study, like in line 284

Answer: We modified the references for different authors throughout the results section.

Location: Lines 237-316.

287: which – typo

Answer: re-phrased.

Location: Lines 290-291.

291: described x2 – repetitive – please revise

Answer: re-phrased.

Location: Lines 292-293.

294: newborns – typo Table 2: heart attack????

Answer: corrected.

Location: Page 15, Table 2.

310: conceptuses – this is not correct – please revise

Answer: corrected.

Location: Page 15 and 16, Table 2.

312: “ showed that the presence of at least one CNS abnormality had the highest prevalence ratio 0.06 (95% 313 CI 0.02-0.09).” Please revise the structure of this sentence

Answer: sentence re-phrased.

Location: Lines 326-328.

315: Is prevalence ratio the best way of expressing this?

Stillbirth – does that mean no studies ad any women who had stillbirths?

IS it not worth joining fetal loss/stillbirth/early or late miscarriage as one category?

Answer: We used the prevalence ratio since it was used in the other systematic reviews, and it allowed us to compare results and perform the meta-analysis of the meta-analysis. We joined miscarriage and stillbirth as one category: fetal loss.

Location: Line 330, Page 19 Fig 6.

368-386 – I don’t really understand why this bit of the discussion. You can’t really compare European data from one period to Brazilian data from another period. Also, this is surely a systematic review looking at the prevalence of microcephaly so why is this talking about other studies not in this SR. The discussion should start off summarising the salient points of the study. This seems like a long ramble into the pathophysiology of transplacental transmission which, although interesting, is missing the point of your SR a little.

Answer: We deleted the European data, maintaining only the data from Brazilian studies. We summarised the salients points of the study. We, with great respect of your point of view, decided to maintain the paragraph about the pathophysiology of transplacental transmission because we believe it is the basis to explain the different outcomes observed in these women.

388: “can be explained mainly” may be best to say: “could be explained by”

Answer: re-phrased.

Location: Line 432.

390: "in addition to the fact that population samples of exposed pregnant women of very different sizes (28 to 19,963 pregnant women)" – please revise structure/grammar

Answer: re-phrased.

Location: Lines 432-439.

393: “Besides, the non-differentiation of the prevalence of microcephaly for exposure to ZIKV in different trimesters may also have contributed to this difference,” Do you mean some studies might have captured more women infected in the first trimester and others more women infected in the third? Please explain further in the text

Answer: The sentence was re-phrased for better comprehension.

Location: Lines 439-441.

396-398: Please revise grammar and structure

Answer: Revised.

Location: Lines 442-444.

400: It would only increase prevalence if these were in utero antenatal surgical procedures which I don’t think they are?

Answer: Sentence deleted.

401-403: Please revise

Answer: re-phrased.

Location: Lines 445-447.

404: Ref 48 is from a textbook so likely out of date. Also – need to compare prevalence in the same population otherwise can’t draw comparisons.

Answer: reference changed, using references of studies from Brazil.

Location: Lines 447-448.

412-413: Please revise

Answer: revised.

Location: LInes 455-457.

414-415: Especially in newborns/infants? Surely this is the only category you think about congenital infections in?

Answer: re-phrased.

Location: Lines 459-465.

417: “intracranial calcification occurs in 30 to 90% of cases of congenital cytomegalovirus (CMV) infection and 50 to 80% of cases of congenital toxoplasmosis” This seems very high – are you sure this is correct? Which country is this data from?

Answer: Information deleted.

423-425: Please revise – unable to follow

Answer: re-phrased.

Location: Lines 468-471.

426: Adolescence?? With congenital ZIKV?

Answer: Information deleted.

463-465: It seem strange to quote one paper and then say it is in agreement with your SR? Surely this paper should be included in the SR?

Answer: sentence modified.

Location: Lines 505-506.

467: Disorders in concepts – please revise, concepts not correct term here

Answer: modified.

Location: Lines 410-412.

466: typo – analyses. Also – if meta-analyses already exist, these should have been incorporated into this meta-analysis using appropriate techniques to make it a more all-encompassing review

Answer: we incorporated the SRs in our study, performing a meta-analysis of the meta-analysis as suggested previously.

Location: Page 12, Table 1. Lines 237 -238; lines 378-385. Page 20, Fig 11.

471-481: please review grammar and structure

Answer: re-phrased.

Location: Lines 531-538.

481: Different outcomes or different definitions or defining criteria?

Answer: different criteria, re-phrased.

Location: Line 543.

485: repetition, study, studies – please revise

Answer: re-phrased.

Location: Lines 545.

487-490: Please revise gramar

Answer: re-phrased.

Location: Lines 547-551.

495: “disorders in fetuses/newborns” – again, be more specific

Answer: re-phrased.

Location: Lines 556-557.

496-498 and conclusion as a whole: Please revise grammar. You have not summarised here what your findings were. These are quite generic comments. It would be good to have closing comments based on your findings

Answer: Conclusions revised and modified.

Location: Lines 556-569.

 

Reviewer #2: 

Thank you very much for each comment. Your insightful suggestions immensely helped to address critical unresolved issues to improve the manuscript ultimately.

The authors performed in December 2019 a well planned and executed meta-analysis of pregnancy and neonatal outcomes associated with maternal Zika virus infection during pregnancy. Among 15 cohorts included in the systematic review, they were able to analyze miscarriage , stillbirths, microcephaly, CNS abnormalities, intracranial calcifications and ventriculomegaly. Due to the heterogeneity of the data, not all cohorts contributed to all endpoint meta-analysis. The manuscript is technically sound, well written and adequately approach the findings in a comprehensive manner in the discussion. The overall findings had already been known in the literature. A missing point that might bring more light into the issue would to characterize what is now generally accepted as typical congenital ZIKV syndrome findings . Among those carrying CNS abnormalities/microcephaly what would be the prevalence rate of severe typical CNS involvement? these data might be available in some of the selected publications. If so, it could be analyzed in the manuscript. Also, it would be helpful to know if other secondary outcomes such as eye, ear abnormalities have been searched for.

Answer: Thank you for the comments. We expanded our search until November 2020, including new outcomes that were not explored in other systematic reviews, such as small for gestational age, low birth weight, prematurity, fetal loss and the positivity if RT-PCR from newborns exposed to ZIKV during their mother's pregnancy.

We re-searched about the different congenital abnormalities in CNS, besides eyes and congenital ear anomalies, trying to characterize a more specific syndrome of Congenital Zika Infection. Unfortunately, there is a lack of this information in most of the cohorts published. We observed the high prevalence of intracranial calcifications and ventriculomegaly among CNS congenital abnormalities. Other CNS congenital abnormalities are cited. So we made a table with all these anomalies described in order of frequency, trying to reach a more specific view of the syndrome. We added a table with the different congenital eyes abnormalities as well. Ear congenital abnormalities were not specified in the cohorts, but we were able to estimate its prevalence. 

Location: Location: Lines 166-167; lines 187-189; lines 292-293; lines 312-316; lines 337-338, Lines 507-530. Table 1, 2 and 3. Figs 7-10.

---

## [Decision Letter · Decision Letter 1]

29 Dec 2020

PONE-D-20-27361R1

Fetal, neonatal, and infant outcomes associated with maternal Zika virus infection during pregnancy: a systematic review and meta-analysis.

PLOS ONE

Dear Dr. Martins,

Thank you for submitting your manuscript to PLOS ONE. After careful consideration, we feel that it has merit but does not fully meet PLOS ONE’s publication criteria as it currently stands. Therefore, we invite you to submit a revised version of the manuscript that addresses the points raised during the review process.

There are remaining comments that need to be addressed. A copyeditor must be contacted by the authors to improve the English grammar and syntax.

We look forward to receiving your revised manuscript.

Kind regards,

Frank T. Spradley

Academic Editor

PLOS ONE

Reviewers' comments:

Reviewer's Responses to Questions

**Comments to the Author**

1. If the authors have adequately addressed your comments raised in a previous round of review and you feel that this manuscript is now acceptable for publication, you may indicate that here to bypass the “Comments to the Author” section, enter your conflict of interest statement in the “Confidential to Editor” section, and submit your "Accept" recommendation.

Reviewer #2: All comments have been addressed

2. Is the manuscript technically sound, and do the data support the conclusions?

Reviewer #2: Yes

3. Has the statistical analysis been performed appropriately and rigorously? 

Reviewer #2: I Don't Know

4. Have the authors made all data underlying the findings in their manuscript fully available?

Reviewer #2: Yes

5. Is the manuscript presented in an intelligible fashion and written in standard English?

Reviewer #2: Yes

6. Review Comments to the Author

Reviewer #2: The authors addressed the referees' questions. By increasing one year of searching for available articles and including other outcomes of interest, they strengthened their work's usefulness to the audience. However, considering that this analysis should have englobed all articles by November 2020, I missed one that complies with the selection criteria published in September, 2020 which could add information to this metanalysis.

"Coutinho CM, et al; NATZIG Cohort Study Team. Early maternal Zika infection predicts severe neonatal neurological damage: results from the prospective Natural History of Zika Virus Infection in Gestation cohort study. BJOG. 2020 Sep 13. doi: 10.1111/1471-0528.16490. Epub ahead of print. PMID: 32920998."

A revision of the English language would additionally improve the manuscript.

7. PLOS authors have the option to publish the peer review history of their article (what does this mean?). If published, this will include your full peer review and any attached files.

Reviewer #2: **Yes: **Marisa Marcia Mussi-Pinhata

---

## [Author Response · Author response to Decision Letter 1]

20 Jan 2021

Once again, we would like to express our full appreciation for allowing us to revise our manuscript entitled " Fetal, neonatal, and infant outcomes associated with maternal Zika virus infection during pregnancy: a systematic review and meta-analysis ". We believe that the inclusion of one more cohort, previously not found in our research, has helped to improve the manuscript ultimately. All the changes in the paper are highlighted in "red" in the revised version (Revised Manuscript Track Changes). We also tried to improve our english in the manuscript.I remain at your disposal to clarify any pending point.

---

## [Editor Report · Decision Letter 2]

25 Jan 2021

Fetal, neonatal, and infant outcomes associated with maternal Zika virus infection during pregnancy: a systematic review and meta-analysis.

PONE-D-20-27361R2

Dear Dr. Martins,

We’re pleased to inform you that your manuscript has been judged scientifically suitable for publication and will be formally accepted for publication once it meets all outstanding technical requirements.

Kind regards,

Frank T. Spradley

Academic Editor

PLOS ONE

---

## [Editor Report · Acceptance letter]

3 Feb 2021

PONE-D-20-27361R2 

Fetal, neonatal, and infant outcomes associated with maternal Zika virus infection during pregnancy: a systematic review and meta-analysis. 

Dear Dr. Martins:

I'm pleased to inform you that your manuscript has been deemed suitable for publication in PLOS ONE. Congratulations! Your manuscript is now with our production department. 

Kind regards, 

on behalf of

Dr. Frank T. Spradley 

Academic Editor

PLOS ONE